# A computational interactome and functional annotation for the human proteome

**José Ignacio Garzón[1], Lei Deng[1,2], Diana Murray[1], Sagi Shapira[1,3], Donald Petrey[1,4], Barry Honig[1,4,5,6,7]\***

[1]Center for Computational Biology and Bioinformatics, Department of Systems Biology, Columbia University, New York, United States; [2]School of Software, Central South University, Changsha, China; [3]Department of Microbiology and Immunology, Columbia University, New York, United States; [4]Howard Hughes Medical Institute, Columbia University, New York, United States; [5]Department of Biochemistry and Molecular Biophysics, Columbia University, New York, United States; [6]Department of Medicine, Columbia University, New York, United States; [7]Zuckerman Mind Brain Behavior Institute, Columbia University, New York, United States

**Abstract** We present a database, PrePPI (Predicting Protein-Protein Interactions), of more than 1.35 million predicted protein-protein interactions (PPIs). Of these at least 127,000 are expected to constitute direct physical interactions although the actual number may be much larger (~500,000). The current PrePPI, which contains predicted interactions for about 85% of the human proteome, is related to an earlier version but is based on additional sources of interaction evidence and is far larger in scope. The use of structural relationships allows PrePPI to infer numerous previously unreported interactions. PrePPI has been subjected to a series of validation tests including reproducing known interactions, recapitulating multi-protein complexes, analysis of disease associated SNPs, and identifying functional relationships between interacting proteins. We show, using Gene Set Enrichment Analysis (GSEA), that predicted interaction partners can be used to annotate a protein's function. We provide annotations for most human proteins, including many annotated as having unknown function.

**\*For correspondence:** bh6@columbia.edu

**Competing interests:** The authors declare that no competing interests exist.

## Introduction

Characterizing protein-protein interaction (PPI) networks on a genomic scale is a goal of unquestioned value (*Ideker and Krogan, 2012*) but it also constitutes a challenge of major proportions. PPIs can be classified as: (1) direct, when two proteins interact through physical contacts; (2) indirect, when the two proteins bind to the same molecules but are not themselves in physical contact; and (3) functional, when two proteins are functionally related but do not interact directly or indirectly – for example, if they are in the same pathway. Considerable experimental effort has been devoted to identifying and cataloging PPIs, but achieving complete coverage remains difficult (*Hart et al., 2006*). For direct and indirect interactions, high-throughput (HT) techniques such as two-hybrid approaches (*Uetz et al., 2000*) and application of mass spectrometry (MS) (*Gavin et al., 2002*; *Havugimana et al., 2012*) are currently the most widely-used approaches for identifying PPIs on a large scale. The PPIs identified by these methods have been deposited in a range of databases (*Ruepp et al., 2010*; *Stark et al., 2011*; *Licata et al., 2012*) but are dominated by interactions

supported by only one experiment, and can be of uncertain reliability (*Sprinzak et al., 2003*; *Venkatesan et al., 2009*; *Rolland et al., 2014*). Moreover, even recent state-of-the-art applications of two-hybrid (*Rolland et al., 2014*) and MS-based approaches (*Havugimana et al., 2012*; *Hein et al., 2015*; *Huttlin et al., 2015*; *Wan et al., 2015*), using protocols involving multiple experiments to ensure quality control, provide interactions for only one-quarter to one-third of the human proteome.

Computational approaches offer another means of inferring PPIs (*Shoemaker and Panchenko, 2007*; *Plewczyński and Ginalski, 2009*; *Jessulat et al., 2011*). Automatic literature curation has produced large databases such as STRING (*Franceschini et al., 2013*), but necessarily these are primarily comprised of proteins whose biological importance is already known (*Rolland et al., 2014*). However, curated interactions are often leveraged to predict additional PPIs using evidence such as sequence similarity (*Walhout et al., 2000*; *Sprinzak and Margalit, 2001*), evolutionary history (*Lichtarge and sowa, 2002*; *Lewis et al., 2010*; *de Juan et al., 2013*), expression profiles (*Jansen et al., 2002*; *Bhardwaj and Lu, 2005*), and genomic location (*Marcotte et al., 1999*). For example, HumanNet (*Lee et al., 2011*) contains 450,000 PPIs for about 15,500 human proteins of which about 400,000 correspond to computational predictions. STRING (*Franceschini et al., 2013*) contains 311,000 interactions based primarily on literature curation for about 15,000 human proteins. These large computational databases contain evidence for all three categories of PPIs. On the other hand, they do not exploit structural information which is perhaps the strongest evidence for a direct physical interaction.

We recently developed a PPI prediction method (*Zhang et al., 2012*), PrePPI, which combines structural information with different sources of non-structural information. In contrast to structure-based approaches that depend on detailed modeling or binding mode sampling (*Lu et al., 2002*; *Tuncbag et al., 2011*; *Mosca et al., 2013*), we developed a low-resolution method that enabled the use of protein structural information on a genome-wide scale. This involved the recognition that homology models of individual proteins and only partially accurate depictions of protein-protein interfaces can contain useful clues as to the presence of an interaction, especially when framed in a statistical context and when combined with other sources of information. Applied to the human proteome, the resulting database (*Zhang et al., 2012*) performs comparably to HT approaches and provides about 300,000 'reliable' interactions for ~13,300 proteins (see below). PrePPI relies heavily on the 'structural Blast' approach (*Dey et al., 2013*), where structural alignments between proteins can detect functional relationships that are undetectable with sequence information alone. The use of structural information in PrePPI is largely responsible for the prediction of a large number of PPIs not inferred with other approaches.

Here, we describe an updated version of PrePPI for the human proteome that incorporates additional interaction evidence. As illustrated in *Figure 1*, the evidence is based on methodological advances in the use of structure, orthology, RNA expression profiles, and predicted interactions between structured domains and peptides (*Chen et al., 2015*). The resulting PrePPI database contains over 1.35 million reliably predicted interactions with considerably increased accuracy compared to the previous version. Moreover, reliable predictions are made for ~17,200 human proteins, about 85% of the human proteome. In addition to the validations reported previously, the new version is subjected to further tests that demonstrate its ability to detect functional relationships for predicted PPIs. Further, by combining PrePPI and gene set enrichment analysis (GSEA), essentially all of the human proteome—including 2000 proteins of unknown function—can be annotated based on the known functions of their predicted interactors.

## Results

### Overview of the PrePPI algorithm

To determine whether two given proteins, A and B, are involved in a PPI, we calculate a set of scores (*Figure 1*) reflecting different types of evidence for a potential interaction. In the previous version of PrePPI (see [*Zhang et al., 2012*] for details), this included:

1. Structural Modeling (SM). A set of coarse interaction models for the direct interaction of A and B is created by superposing models or structures of A and B on experimentally determined structures of the physical interaction between proteins A' and B'. These models are scored

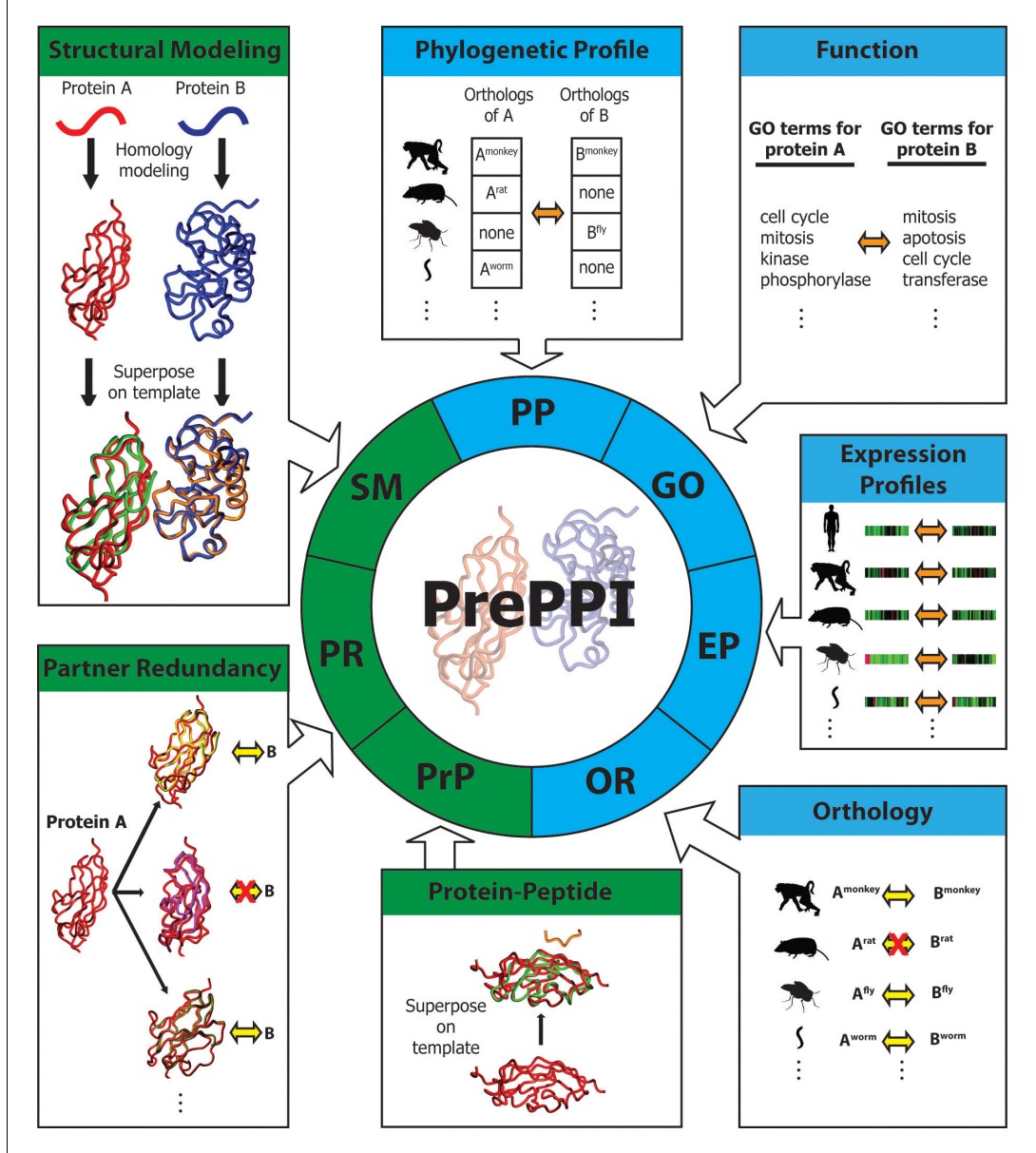

**Figure 1.** Overview of PrePPI prediction evidence and methodology. Each box represents a type of interaction evidence used in PrePPI. Those titled with a green background use structural information and those with a blue background do not. Contained in each box is a visual outline of how that type of evidence is used (details are provided in the text and Materials and methods). Overall, an interaction between two given proteins A and B is inferred based on some similarity to others that are known to interact (yellow double arrows) or if they have other properties that correlate in some way (orange double arrows). A red 'X' indicates that an interaction between two proteins does not exist.

based on the structural similarity of A to A' and of B to B' and the properties of the modeled interface between A and B.

2. Phylogenetic Profile (PP). This score indicates the extent to which orthologs of A and B are present in the same species (*Pellegrini, 2012*).

3. Gene Ontology (GO). This signifies whether A and B have similar functions based on their proximity in the GO hierarchy (*Ashburner et al., 2000*).

Additional sources of evidence incorporated into the current version of PrePPI are as follows (see Materials and methods for details).

4. Orthology (OR). This measures the likelihood that A and B interact if their orthologs A' and B' in other species interact. To provide the broadest coverage of the human proteome, we use a novel approach to combine information from multiple orthology databases. Although orthology is also used in the phylogenetic profile (PP) term, we note that the new OR term directly reflects the fact that orthologs of A and B may interact in another species whereas this is not the case for PP. The two sources of evidence are largely complementary (Pearson correlation coefficient=0.003).

5. Expression Profile (EP). We developed a new EP score, which is designed to reflect similarities in the expression patterns of two human proteins A and B. The previous version of PrePPI also used EP evidence, but only for human proteins. In contrast, we now consider not only similarities in expression patterns of A and B, but of orthologs of A and B in a set of 11 model organisms. Again, this term does not correlate with the OR or PP term (correlation coefficients of 0.02 and 0.1 respectively).

6. Partner Redundancy (PR). This score is based on the assumption (*de Chassey et al., 2008*) that if protein B is known to interact with other proteins $P_1, P_2, \ldots P_n$, that are structurally similar to A, the likelihood that B also interacts with A increases with $n$. We also include a score based on the number of times a protein structurally similar to A interacts with a protein structurally similar to B, where the structurally similar proteins can be obtained from any species.

7. Protein Peptide (PrP). This score reflects the likelihood of an interaction between A and B if A contains a structured domain from a given domain family and B contains a short sequence motif similar to one known to interact with that type of structured domain (*Chen et al., 2015*).

A Bayesian approach combines the above scores into a single likelihood ratio (LR) reflecting the probability of a PPI between A and B (*Figure 1*, and *Equation 1* in Materials and methods). The Bayesian network was trained on a positive reference set of 42,636 yeast PPIs compiled from various PPI databases and reported in at least two publications (the yeast HC reference set) and a negative set of all pairs for which there is no literature evidence of an interaction (the yeast N set). The Bayesian network was evaluated with a positive reference set of 26,983 human interactions reported in at least two publications (the human HC reference set) and a negative reference set consisting of 1,632,716 pairs of proteins where each individual protein belongs to a separate cellular compartment as annotated by GO (the human N set; see Materials and methods). A separate evaluation was carried out with the positive reference set (PRS, ~500 human PPIs) and random reference set (RRS, ~700 pairs of human proteins) assembled in *Rolland et al. (2014)*.

## Overall prediction performance

The receiver operating characteristic (ROC) curves in *Figure 2a* (and see *Figure 2—source data 1*) assess the performance of the new and previous versions of PrePPI on the human HC and N reference sets. The addition of new evidence considerably increases the true positive rate at low false positive rates (solid black and gray curves). As seen by comparing the dashed black and grey curves, a similar PrePPI performance is observed based on the PRS and the RRS from *Rolland et al. (2014)*, providing support for the robustness of our PrePPI validation. *Figure 2a* also displays the performance of two state-of-the-art high-throughput databases, here termed Y2H (*Rolland et al., 2014*), and BioPlex (*Huttlin et al., 2015*), both evaluated on the PRS and RRS. As can be seen, at a similar false positive rate (FPR), PrePPI performance is comparable to that of these two databases.

*Figure 2b* plots the percentage of interactions PrePPI recovers in PRS and RRS as a function of LR. We previously used an LR cutoff of 600 to define 'reliable' predictions based on the argument (*Jansen et al., 2003*) that this value corresponds to a 'real' false positive rate of 0.5 (i.e., the false positive rate that would be expected if every true interaction was known). The fact that PrePPI recovers essentially no RRS interactions for LR>600 provides an independent validation of this definition of a reliable prediction. About 60% of the PRS database is recovered at this LR cutoff while approximately 50% of the human HC reference set is also recovered. *Figure 2b* also shows that the Y2H and Bioplex databases recover ~7% and ~16% of the PRS interactions, respectively.

## The contribution of different sources of information to PrePPI performance

The current PrePPI database contains about 1.35 million reliable PPI predictions, a significant increase over the 300,000 PPIs in the previous version. Some of this increase is due to the availability of new structural information; about 400,000 reliable predictions are made in the current

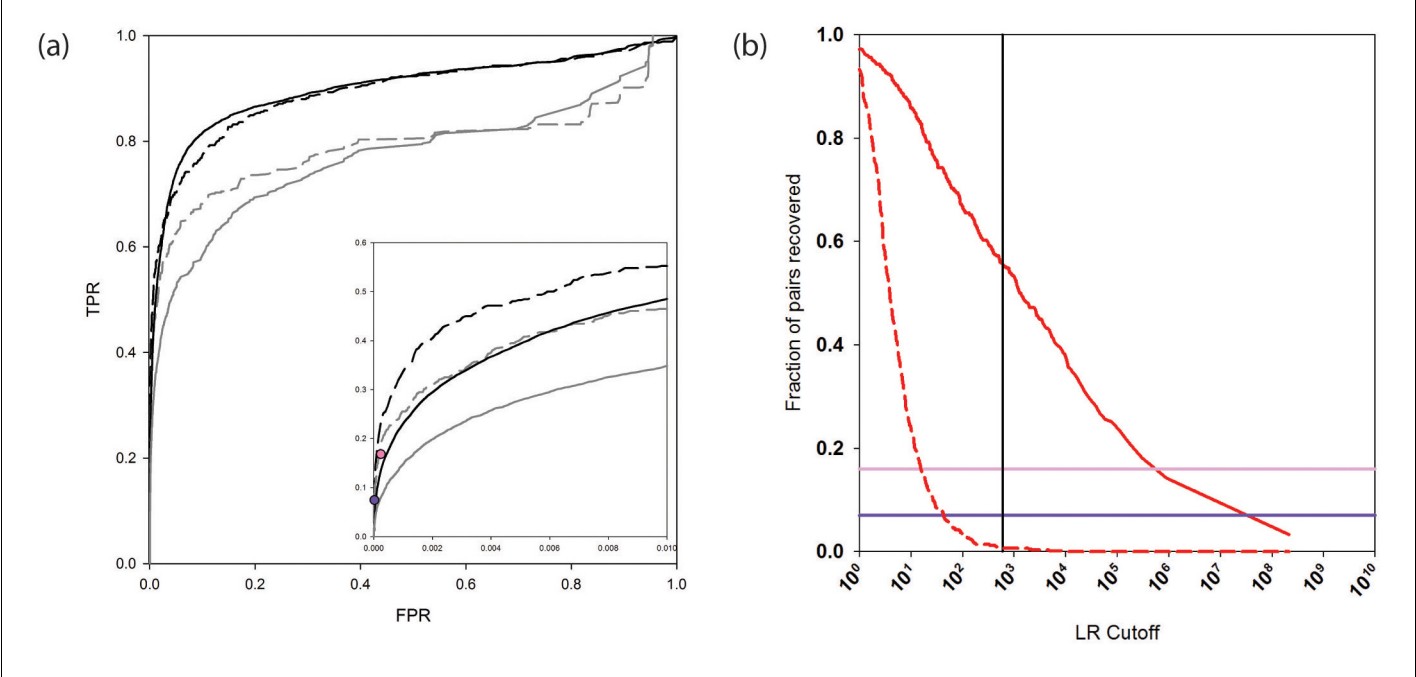

**Figure 2.** PrePPI prediction performance. (**a**) ROC curves assessing PrePPI prediction performance using different reference sets. The y-axis shows the fraction of interactions recovered from a positive reference set (TPR, true positive rate) vs. the fraction recovered from a corresponding negative reference set (FPR, false positive rate). The solid grey and black lines are based on the human HC and N reference sets defined in Materials and methods and compare performance based only on the sources of evidence used in the previous version of PrePPI (grey) and incorporating the new sources of evidence (black). The dashed lines are based on the PRS/RRS reference sets described in the text. The inset shows recovery in the low FPR region. (**b**) Fraction of PRS (solid line) and RRS (dashed line) interactions recovered with an LR above the cutoff shown on the x-axis. Solid horizontal lines show the fraction of the PRS sets recovered by the Y2H (dark purple) and BioPlex (light purple) databases.

The following source data is available for figure 2:

**Source data 1.** PrePPI LRs for interactions in the HC, PRS and RRS reference interaction sets.

implementation if we include the new structural data but not the new sources of evidence. However, most of the increase is due to the new sources of evidence now integrated into PrePPI. The second column in *Table 1* reports the number of predictions with an LR > 600 for each source of evidence taken on its own and also gives numbers obtained from the combination of all structural or non-structural evidence. It is clear that SM is by far the single most important source of evidence, as was also found in our earlier study (*Zhang et al., 2012*). However, if we combine all sources, structural evidence yields about 127,000 reliable predictions while the combination of all non-structural sources yields about 115,000 predictions, so that both are approximately of equivalent importance on their own. That the use of all sources of evidence increases the total number of high confidence predictions to 1.35 million is a clear indication of the value of integrating independent evidence in a Bayesian framework.

The third column of *Table 1* lists numbers of reliable predictions that can be made by *removing* individual sources of evidence. Here again, SM is found to be the most important contributor to the total number of predictions but clearly each source is required to achieve 1.35 million predictions. This is not surprising given the fact that the total LR is a product of six individual terms (*Equation 1*). Notably the removal of the orthology term actually increases the total number of reliable predictions. This is because many PPIs have $LR^{OR} < 1$, which serves to counter the evidence from other sources.

*Figure 3* analyzes the contribution of homology modeling to performance. The ROC curves in the figure are based on predictions where either the structures of both interactors are taken directly from the PDB (red line) or where the interactors can be either homology models or crystal structures

**Table 1.** Contributions of different sources of evidence to PrePPI performance. The second column shows the number of reliable predictions that can be made using only the evidence identified in the first column. The third column shows the number of predictions that can be made without the evidence shown in the first column, but including all other types of evidence. Note that excluding the OR evidence results in more interactions than using all evidence. This is because there are many interactions for which the LR based on orthology is less than 1. Rows with a green background indicate evidence types that use structural information and rows with a blue background evidence types that are based on non-structural information.

| Evidence | Number of interactions from this source alone | Number of interactions from all other sources |
|---|---|---|
| All | 1,354,008 | 0 |
| SM | 48,090 | 675,965 |
| PrP | 1396 | 1,041,951 |
| SM,PrP | 49,457 | 192,315 |
| PR | 0 | 1,022,829 |
| SM,PrP,PR | 127,140 | 114,970 |
| OR | 624 | 1,388,341 |
| PP | 0 | 802,762 |
| CE | 0 | 726,592 |
| GO | 0 | 732,198 |
| OR,PP,CE,GO | 114,970 | 127,140 |

(black line). As can be seen from the ROC curves in the figure, the use of homology models in predicting interaction models leads to a clear improvement in PrePPI performance. However, more significantly, ~250,000 reliable predictions can be made without homology models as compared to 1.24 million with homology models so that models have a quite dramatic effect on recovery. (The difference between the total database of 1.35 million and the number of predictions based only on the experimentally determined structure reflects the ~115,000 predictions in PrePPI that have no contribution from structure – *Table 1*).

## Assessing PrePPI testing and training

In this section we consider possible sources of error in the training and testing procedures used in the development of PrePPI.

### a. Independence of training and test sets

Although the Bayesian network was trained using a reference set of interactions from yeast, information about human interactions could potentially bias the training and testing procedures since there are human orthologs of interacting proteins in yeast. To test for this, we identified the 2671 human PPIs that have orthologous PPIs in the yeast reference set. We recalculated the ROC plots, excluding the evidence based on orthology for these 2671 interactions (i.e., removing $LR^{OR}$ from *Equation 1*). The performance as measured by ROC curves was essentially identical (data not shown). Moreover, of the 13,207 interactions in the human HC reference set that were identified with LR>600 including evidence from orthology, 12,814 also had LR>600 without this evidence, suggesting that evidence from orthology does not artificially influence PrePPI performance.

### b. The role of paralogs and related issues of specificity

Its heavy reliance on structural similarity suggests that PrePPI might assign artificially high scores to interactions between pairs of proteins, both of which are paralogous to proteins in the template. However, only about 13% of the 1.35 million reliable predictions correspond to cases where the proteins involved are paralogs of proteins in the template (as defined by either the Gopher, OMA or

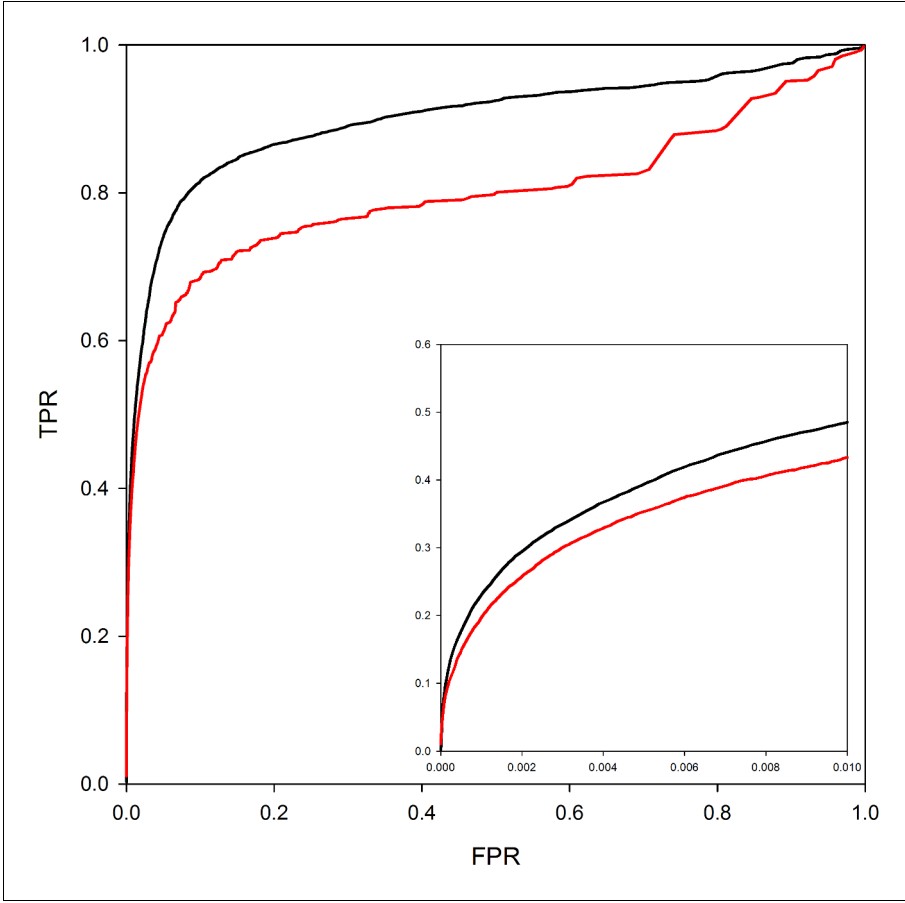

**Figure 3.** PrePPI performance with and without homology models. The red line shows PrePPI performance using only crystal structures available in the PDB for individual proteins. The black line shows the performance if both homology models and crystal structures are used.

KEGG databases). Moreover, only about 25% of the 127,000 predictions derived from structure alone are based on paralogs.

A serious concern however is the question of whether the PrePPI scoring function is able to distinguish among paralogs since they all have similar structures and thus, for example, two families of paralogs might all be expected to interact with one another. For example, can PrePPI specifically associate small GTPases with their corresponding GEFs or GAPs? In our previous paper (*Zhang et al., 2012*) we carried out an analysis of predicted interactions between proteins containing GTPases, GEFs, GAPs, and SH2 and SH3 domains and found a wide range of SM and total PrePPI LR's for interactions between proteins containing these domains. Indeed, the SM scoring function accounts for a significant degree of specificity due in part to the fact that more structurally similar paralogs tend to have similar binding properties and because we score the interaction model based on the propensity of the predicted interface to bind to other proteins. Further discrimination is achieved by non-structural evidence such as co-expression. We have repeated the analysis of specific domain families within the new version and have confirmed our previous findings (data not shown). Here we carry out a more extensive analysis of specificity among paralogs.

*Figure 4a and b* plot the number of modeled interactions as a function of LR for interacting proteins annotated as paralogs to their structural templates (as determined from either the Gopher, OMA or KEGG databases). There is a wide range of LRs, either based on structural modeling alone (*Figure 4a*), or considering all sources of information used in PrePPI (*Figure 4b*). The wide range of LRs obtained from structural modeling demonstrates the sensitivity of our scoring function while the even wider range for the total PrePPI LR shows that other sources of evidence make important

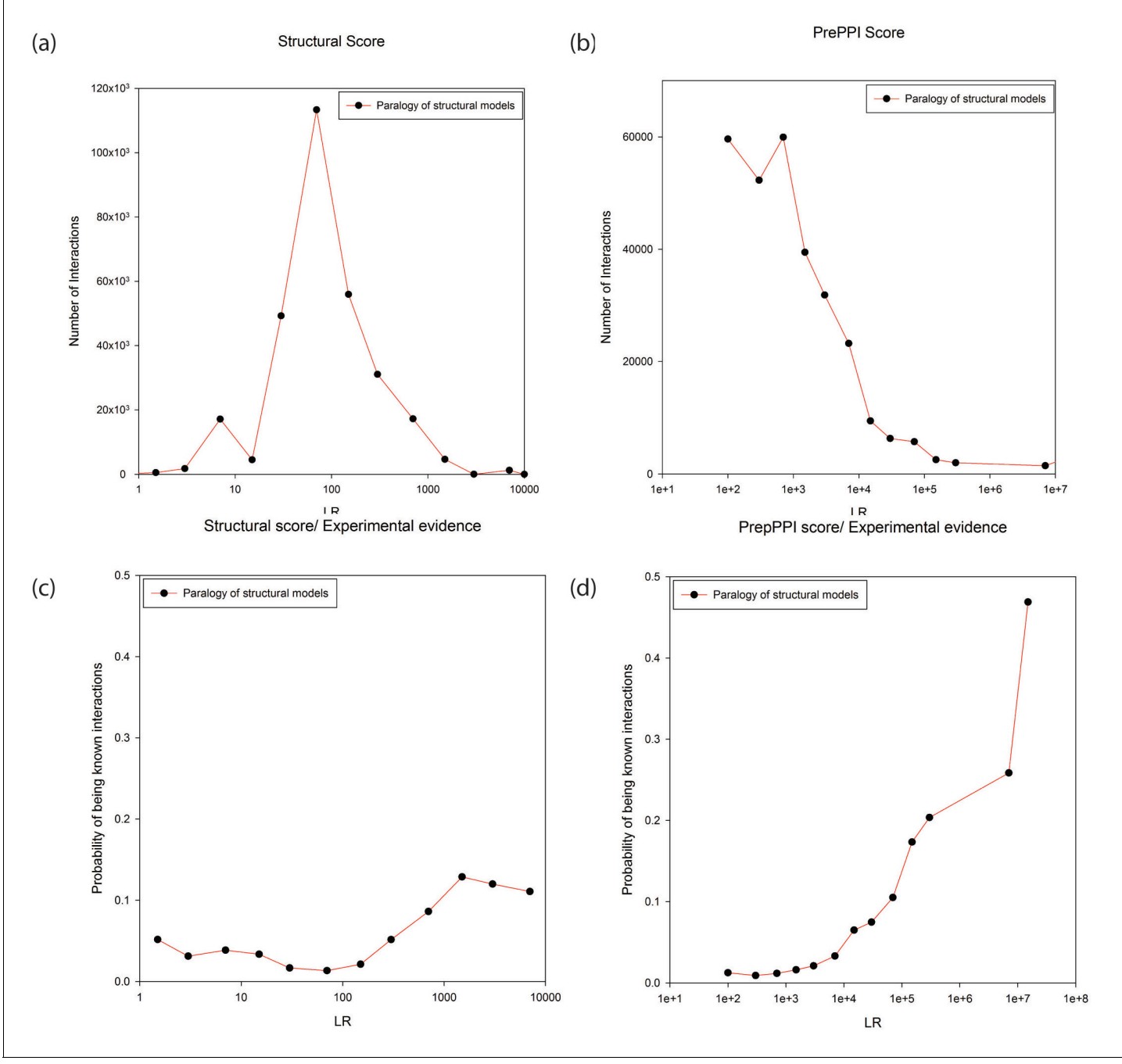

**Figure 4.** Predictions between proteins annotated as paralogs of proteins in the template. Panels (**a**) and (**b**) show the numbers of predictions (y-axis) as a function of LR (x-axis). In Panel (**a**) the LR is based only on the SM term and in Panel (**b**) is based on all sources of evidence. Panel (**c**) and (**d**) show the probability (y-axis) of interactions being in the HC set as a function of LR (x-axis). In Panel (**c**) the LR is based only the SM score and in Panel (**d**) it is based on all sources of evidence.

contributions to specificity. *Figure 4c and d* show the probability that a predicted interaction is true as a function of LR (an interaction is considered true if there is publication reporting it as determined from the databases listed in Materials and methods). Both figures (but especially *Figure 4D*) suggest that there is a significant degree of specificity implicit in the PrePPI scoring function in that the probability of being true increases with LR.

## c. Overlap between the current and previous versions of PrePPI

We compared the set of reliable predictions made in the previous version of PrePPI to the current set of reliable predictions. Overall, about 63% of the interactions assigned an LR>600 in the previous version are also assigned an LR>600 in the current version. Of course any interactions with an LR>600 in the previous version of the database will have some LR value in the current one, but for 37% of the cases the value is now <600. It is possible then that some true interactions are lost in this way. However, of the ~313,000 reliable predictions in the previous version, 3683 are in the human HC reference set and nearly all (3456) are assigned an LR>600 in the current database. Thus, although some true interactions are lost, overall the new evidence and new data incorporated into PrePPI appear to have the desired effect of improving prediction performance. This is due to additional human data, a larger yeast training set, and new sources of evidence introduced in the current version of PrePPI. These results highlight the importance of regular updating of the database and we currently have plans to update annually.

## PrePPI predictions are functionally and clinically relevant

### Shared GO annotation

We calculated the percentage of reliably predicted PPIs for which two interacting proteins are annotated with the same GO 'biological process' (BP). We considered GO BP terms at the second level (see Materials and methods for details). After removing the Gene Ontology (GO) term from the PrePPI LR (LR$_{GO}$ in *Equation 1*), the number of PPIs predicted by PrePPI decreases to about 732,000 (*Table 1*), and 89% of these share a second level BP term. This percentage is less than that found in STRING (95%) and in the human HC set (95%). However, since the human HC reference set is comprised of high confidence experimentally determined and literature-based interactions, which are used in GO annotations, it is encouraging that PrePPI yields comparable results. This indicates that PrePPI predictions can be used to infer functional relationships. For comparison, 55% of a random set of ~700,000 protein pairs share a GO BP term. *Figure 5a* breaks down these GO results for fifteen of the level-two biological process terms. For these terms, the human HC reference (purple bars, *Figure 5—source data 1*) recapitulates the highest percentages of pairs. PrePPI (green, *Figure 5—source data 2*) and STRING (pink, *Figure 5—source data 3*) exhibit similar performance, and all three differ significantly from the random set (black).

### Multi-subunit complexes

We tested the ability of PrePPI to identify indirect interactions by determining the extent to which members of complexes compiled in CORUM, a database of ~1800 multi-subunit complexes (*Ruepp et al., 2010*), are recapitulated. PrePPI LRs between every pair of proteins in a CORUM complex are tabulated, and the largest LR at which all members of the complex form a connected graph is recorded. *Figure 5b* plots the fraction of CORUM complexes recovered as a function of LR (and see *Figure 5—source data 4*). Approximately 2/3 of CORUM complexes are fully recovered at an LR cutoff of 600 (red curve). To verify that our high-recovery rate is not simply an artifact of the large number of PrePPI predictions, this analysis was repeated after replacing all subunits of each CORUM complex with an equivalent number of randomly chosen proteins. Essentially no randomly constructed complexes are recovered for an LR cutoff > 600 (red dashed curve), again validating the reliability of this cutoff. Notably, due to their smaller sizes, the Y2H and BioPlex datasets recover ~5% (dark purple line) and ~20% (light purple line) of the CORUM complexes, respectively.

The ability to account *a posteriori* for members of a complex does not imply the ability to predict its subunit composition. As a step towards developing a predictive algorithm, we randomly chose two members of each of the 1258 CORUM complexes that contain more than 2 subunits. We then asked whether all members of a given complex could be identified from PrePPI predicted interaction partners of the randomly chosen pair with LR > 600. This was possible for 865 of 1258 complexes that have three or more subunits. All members of a complex were identified within the top 10 partners for 305 of 892 complexes with three to six subunits, and for 601 of these complexes, all members were identified within the top 100 partners. Moreover, all members could be identified in the top 100 predicted interaction partners for 43 of 339 complexes with seven to 22 subunits. The results of this and the preceding paragraph suggest that the PrePPI database in conjunction with

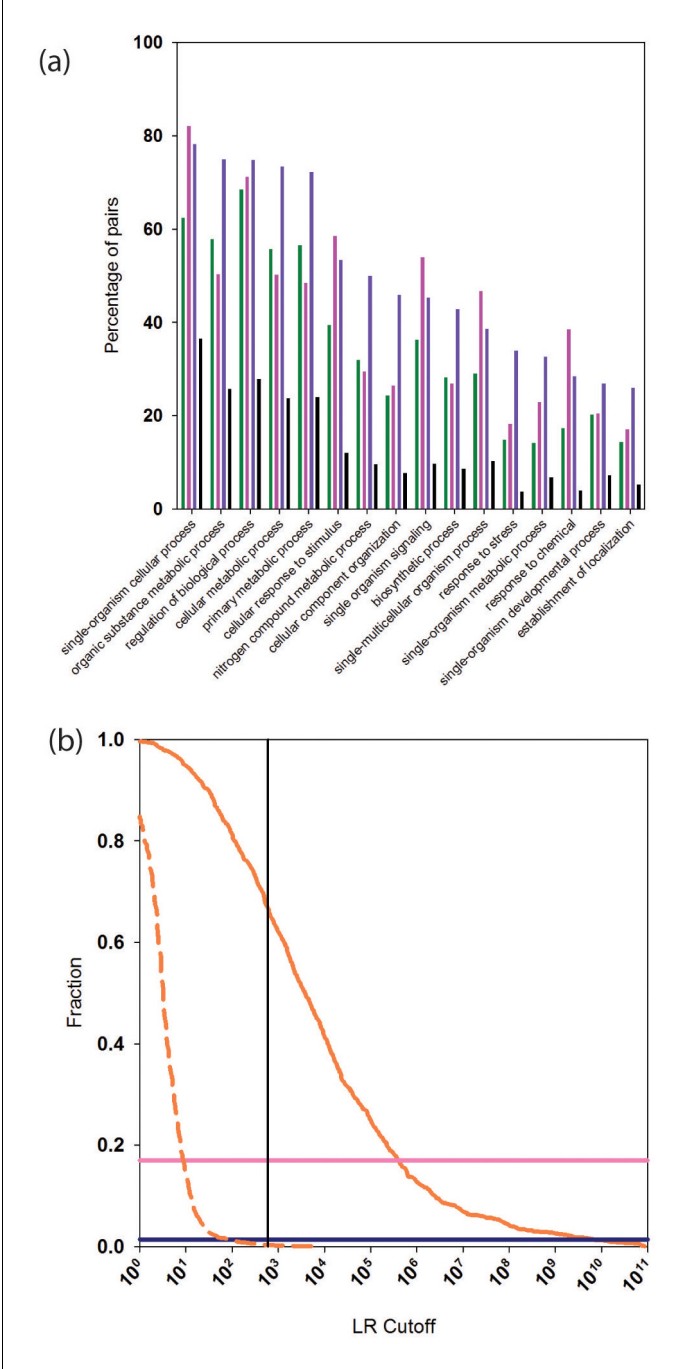

**Figure 5.** Functional relationships of PrePPI predicted interaction partners. (**a**) Percentage of pairs of proteins (y-axis) that are predicted to interact and that share the GO biological process term on the x-axis. Green bars are for PrePPI predictions, pink for the STRING database, purple for the human HC reference set, and black bars are for a random set of protein pairs. (**b**) The solid orange line shows the fraction of CORUM complexes (y-axis) recovered as a function of LR (x-axis). The dashed line shows the recovery of randomly generated complexes with the same number of subunits as those in CORUM. The horizontal lines show the percentage of CORUM complexes recovered by the Y2H (dark purple) and BioPlex (light purple) interaction sets. The vertical line shows the point on the x-axis with LR=600.

The following source data is available for figure 5:

**Source data 1.** List of interactions in the HC reference set that share a GO biological process term at the second level.

*Figure 5 continued on next page*

*Figure 5 continued*

**Source data 2.** List of PrePPI predicted interactions that share a GO biological process term at the second level.

**Source data 3.** List of interactions in the STRING database that share a GO biological process term at the second level.

**Source data 4.** List of CORUM complexes recovered by PrePPI.

graph and network algorithms should make it possible to predict members of multi-protein complexes.

## Disease phenotypes

ClinVar aggregates information about genomic variation and its relationship to human health through a database of 2700 proteins with non-synonymous single nucleotide polymorphisms (SNPs) annotated as associated with specific diseases (*Landrum et al., 2014*). From these 2700 proteins, we found 3714 pairs for which both proteins are associated with the same disease and denote this as the ClinVar positive set. While most diseases are polygenic, it is not unreasonable to assume that a subset of the proteins found to have genomic variants implicated in the same disease may interact either directly or functionally. PrePPI predicts interactions for 854 of these 3,714 disease-similar pairs (see *Supplementary file 1*). To test the significance of this finding, we compiled two negative sets: (1) 3714 ClinVar pairs randomly chosen subject to the constraint that each protein in a pair is associated with a different disease, and (2) 3714 randomly chosen protein pairs from the 1000 Genomes Project subject to the constraint that the proteins contain SNPs predicted to be benign (The Genomes Project 2015). An order of magnitude fewer pairs in these negative sets are predicted by PrePPI: 85 and 34 pairs, respectively. The success of PrePPI in accounting for disease-related PPIs suggests the predicted PPIs should help uncover broader functional interaction networks associated with specific diseases, especially for cases in which only one disease-associated protein may be known.

We examined the ability of other databases to predict pairs of disease-similar proteins. The Y2H and BioPlex sets, respectively, identify 5 and 63 pairs of the ClinVar positive set while the union of four computationally derived databases–STRING, PIP (*McDowall et al., 2009*), I2D (*Brown and Jurisica, 2007*) and HumanNet–predict 764 ClinVar positive pairs. Notably, of the 854 that PrePPI predicts, 617 do not appear in databases of experimentally determined or literature-curated human interactions (see Materials and methods for the databases examined), and 326 do not appear in the other computationally derived databases. The results obtained from the analyses of shared GO annotation, CORUM complexes, and ClinVar disease-similar proteins demonstrate that PrePPI can be used to identify biomedically meaningful interactions, many of which are unique.

The added dimension of protein structure inherent in PrePPI makes it possible to go a step further than just identifying PPIs. Specifically, if a disease-associated single nucleotide polymorphism (SNP) is present in a PPI interface, disruption of the relevant interaction is potentially responsible for the disease phenotype. If this occurs frequently, it would be expected that disease-associated SNPs would be enriched in interfaces. To test this, we compiled disease-associated SNPs from ClinVar and The 1000 Genomes Project and found that the odds ratio for such SNPs to be in PrePPI predicted interfaces was 1.6. This is statistically significant, calculated as in (*Wang et al., 2012*) and similar to the results obtained in that study. We extend the analysis here to a larger set of SNPs and many predicted interactions not currently available in databases. We also analyzed a set of SNPs annotated as benign in the 1000 Genomes Project. Benign SNPs are significantly underrepresented in interfaces (odds ratio of 0.5). Taken together, these results add to the growing evidence for the importance of PPIs in disease (*Wang et al., 2012*, *Kamburov et al., 2015*, *Porta-Pardo et al., 2015*) and demonstrate that the PrePPI-predicted interfaces are accurate enough to identify disease-related mutations.

## Functional annotation using gene set enrichment analysis (GSEA)

Gene set enrichment is a method to identify genes/proteins associated with a given property (e.g., annotation or phenotype) that are overrepresented in a given collection of genes/proteins. A type of GSEA (*Subramanian et al., 2005*), originally developed for RNA expression analysis, takes as input a list of proteins ranked according to each protein's differential expression under different conditions, and outputs predefined gene sets containing proteins that are enriched at the top of this ranked list. If enriched gene sets are defined in terms of GO annotations, they provide a functional profile associated with the differential expression profile. We modified the approach as outlined in *Figure 6A*. For each protein, Q, in the human proteome, we construct a list of all other human proteins ranked according to the PrePPI LR for their interaction with Q. A potential functional annotation for Q is then inferred from gene sets enriched at the top of this ranked list. To test for accuracy, we carried out this procedure for the 10,800 human proteins that (1) are annotated with a GO term and (2) have one or more gene sets associated with that GO term in the Molecular Signatures Database, mSigDB (*Subramanian et al., 2005*). As above, because this test involves identification of GO-related annotations, $LR_{GO}$ is excluded from the full PrePPI LR in *Equation 1*. (However, for applications with other gene sets, this term would be included.)

*Figure 6B* illustrates that the top-ranked gene sets in the PrePPI predicted interactors of Q contain an accurate reflection of that protein's function. Specifically, for ~2100 of the 10,800 query proteins Q, the most enriched gene set is associated with a correct biological process (BP), cellular compartment (CC), or molecular function (MF) GO term for Q. For ~5500 proteins, a correct GO annotation is obtained within the ten most enriched gene sets. *Figure 6C* illustrates two examples: (1) Well-known functions of the tumor suppressor BRCA2, including its involvement in DNA repair, nuclear localization, and cell-cycle control, are correctly identified; in addition, the procedure identifies the recently discovered role of BRCA2 in cytoskeletal organization (*Niwa et al., 2009*). (2) PEX1, the peroxisomal biogenesis factor 1, is the only protein associated with 'Zellwegers syndrome' in ClinVar; the procedure correctly identifies known functions of PEX1, including its association with the peroxisome and its classification as an ATPase.

We applied the procedure outlined in *Figure 6A* to the full human proteome and found enriched gene sets for all but six proteins, where we define 'enriched' as q-value<0.01 as reported by the

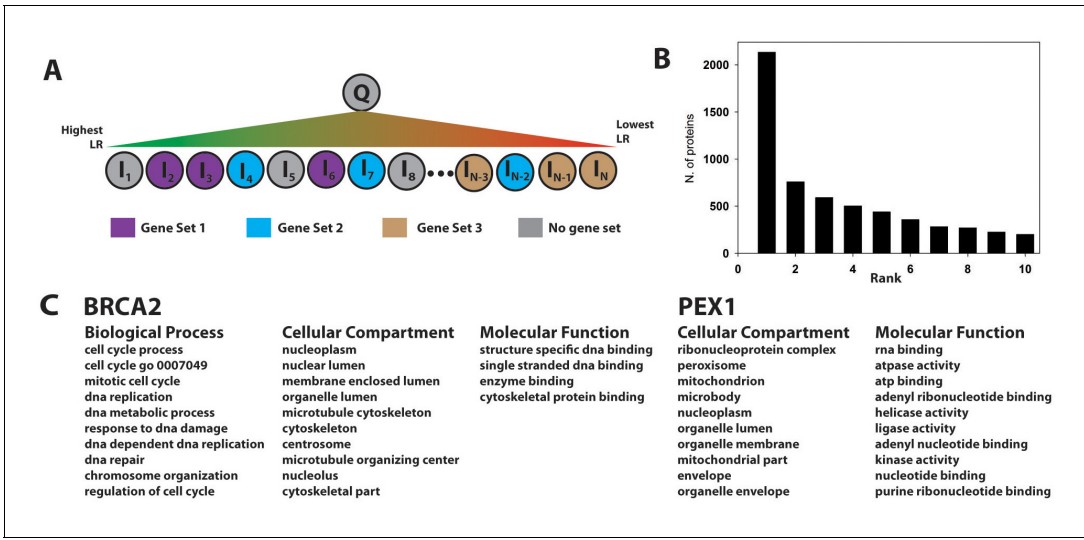

**Figure 6.** Gene set enrichment using the PrePPI interactome. (**A**) To infer a function for a given protein, Q, all proteins in the human proteome, Ii, are placed in a list and sorted according to the interaction LR between Ii and Q. This list is then searched for gene sets associated with a given GO annotation enriched among the high-scoring interactors of Q. In the example in panel **A**, Gene Set 1 would be enriched whereas Gene Sets 2 and 3 would not be, since the proteins in those sets are either evenly distributed throughout the ranked list or clustered with proteins that are unlikely to interact with Q. (**B**) Histogram showing the position, in the list of enriched gene sets, of the first set associated with a known GO annotation for Q. Gene sets are ranked according to their enrichment at the top of LR-ranked list of interactors {Ii} of Q. (**C**) Top ranked gene sets found for two examples, BRCA2 and PEX1.

mSigDB software (*Subramanian et al., 2005*). The average q-value for the top 10 enriched gene sets was highly significant (0.0002). The GO annotations associated with each gene set are available from the PrePPI database for each protein (*Zhang et al., 2013*) including ~2000 proteins of unknown function (PUFs). GSEA-derived annotations for two examples are described below.

FAM69 consists of five members: one of the annotated functions (Q8NDZ4: DIA1, Deleted in autism protein 1, C3orf58) (*Beigi et al., 2013*) and four of unknown function (Q5T7M9: FAM69A; Q5VUD6: FAM69B; Q0P6D2: FAM69C; Q9H7Y0: CXorf36). These proteins were previously characterized as cysteine-rich transmembrane, ER-associated proteins of unknown function (*Tennant-Eyles et al., 2011*). More recently, a computational bioinformatics analysis predicted that they belong to the Protein Kinase-like Clan (*Dudkiewicz et al., 2013*). The GSEA/PrePPI procedure provides a consistent annotation for the four PUFs (and the entire FAM69 family). First, models for the interactions between the FAM69 PUFs and their gene set-enriched interactors are based on template structures of biologically relevant homodimers of protein kinases. Second, some of the GO terms associated with these interactors include 'protein amino acid phosphorylation' and 'enzyme-linked receptor signaling pathway' for biological process, and 'protein kinase activity, 'identical protein binding,' and 'transmembrane receptor protein kinase activity' for molecular function. Taken together, these clues predict, in a completely automatic way, a more comprehensive annotation than that previously provided experimentally or computationally.

In a second example, the procedure provides a novel annotation for the protein FAM178B (Q8IXR5). FAM178A, now known as SLF2 (SMC5-SMC6 complex localization factor protein 2), was recently shown to be involved in DNA repair (*Dudkiewicz et al., 2013*). Consistent with this finding, five of the top interactors for FAM178B are annotated with the GO molecular function 'transcription factor activity.' Examination of these interactors suggests that FAM178B interacts with transcription factors (e.g. STAT1, STAT3, and ESR2).

## Discussion

### Overlap between databases

As delineated in *Table 2*, PrePPI contains 1.35 million entries, more than any other computationally or experimentally derived interaction database of human interactions. PrePPI's size is due in part to the fact that it contains both direct and indirect interactions but this is also the case for computationally derived databases such as HumanNet, STRING and Ophid. Notably, the inclusion of indirect interactions (i.e. functional or those mediated by another molecule) is particularly helpful in the GSEA analysis since it provides a much larger pool of 'interactors' to be used in function annotation. *Table 2* lists the number of PPIs in a number of widely used computational and high-throughput databases and the overlap between them. The entry 'Computational All' represents the union of other current computational databases (see *Table 3* for the number of interactions uniquely contained in each). A number of features emerge from the data summarized in *Tables 2* and *3.* First, of

**Table 2.** Interaction database overlap. Each cell shows the number of interactions shared by the databases indicated in blue at the top and right. Green boxes along the diagonal show the total number of interactions in a single database.

| PrePPI | Y2H | BioPlex | PIP | I2D Ophid | HumanNet | String | Comp. All | Hum.Exp. | |
|---|---|---|---|---|---|---|---|---|---|
| 1,354,007 | 972 | 5364 | 17,639 | 67,556 | 76,905 | 123,457 | 212,463 | 44,864 | PrePPI |
| | 13,584 | 425 | 140 | 13,470 | 804 | 918 | 13,584 | 1777 | Y2H |
| | | 56,553 | 918 | 5689 | 4361 | 5549 | 56,553 | 4763 | BioPlex |
| | | | 44,148 | 6253 | 10,324 | 703 | 44,148 | 5154 | PIP |
| | | | | 296,008 | 56,584 | 53,178 | 296,008 | 160,581 | I2D Ophid |
| | | | | | 458,518 | 58,512 | 458,518 | 44,047 | HumanNet |
| | | | | | | 311,635 | 311,635 | 45,890 | String |
| | | | | | | | 1,004,622 | 162,065 | Comp. All |
| | | | | | | | | 169,368 | Hum. Exp. |

**Table 3.** Unique database interactions. Each cell shows the number of interactions contained in the database indicated on the left in blue, but not in the database indicated at the top in blue. Green boxes show the total number of interactions in a single database

| | PrePPI | Y2H | BioPlex | PIP | I2D Ophid | HumanNet | String | Comp.All | Human Exp. |
|---|---|---|---|---|---|---|---|---|---|
| PrePPI | 1,354,007 | 1,353,035 | 1,348,643 | 1,336,368 | 1,286,451 | 1,277,102 | 1,230,550 | 1,141,544 | 1,309,143 |
| Y2H | 12,612 | 13,584 | 13,159 | 13,444 | 114 | 12,780 | 12,666 | 0 | 11,807 |
| BioPlex | 51,189 | 56,128 | 56,553 | 55,635 | 50,864 | 52,192 | 51,004 | 0 | 51,790 |
| PIP | 26,509 | 44,008 | 43,230 | 44,148 | 37,895 | 33,824 | 36,245 | 0 | 38,994 |
| I2d Ophid | 228,452 | 282,538 | 290,319 | 289,755 | 296,008 | 239,424 | 242,830 | 0 | 135,427 |
| HumanNet | 381,613 | 457,714 | 454,157 | 448,194 | 401,934 | 458,518 | 400,006 | 0 | 414,471 |
| String | 188,178 | 310,717 | 306,086 | 303,732 | 258,457 | 253,123 | 311,635 | 0 | 265,745 |
| Comp. All | 792,159 | 991,038 | 948,069 | 960,474 | 708,614 | 546,104 | 692,987 | 1,004,622 | 842,557 |
| Human Exp. | 124,504 | 167,591 | 164,605 | 164,214 | 8787 | 125,321 | 123,478 | 7303 | 169,368 |

the 2.1 million unique PPIs appearing in some database, 1.1 million are unique to PrePPI, 800,000 are unique to 'Computational All,' and 200,000 are common to PrePPI and 'Computational All.' Regarding the total number of PPIs, the large number identified by PrePPI and other computationally derived databases should not be surprising since it potentially includes any pair of proteins that participate in a shared biological pathway.

A second observation derived from *Tables 2* and *3* is the general lack of overlap between different databases. This feature seems almost universal. PrePPI, given its size, has substantial overlap with other resources: over a third of STRING, 15% of HumanNet, and 25% of Ophid (*Brown and Jurisica, 2007*). PrePPI also overlaps relatively well with the set of interactions that are reported in a range of databases that serve as repositories for literature-curated and high-throughput experimentally determined interactions ('Human Experimental' entry in *Table 2*). Other computational databases also exhibit comparable overlap, but these databases typically include interactions available in experimental databases. It is thus striking that PrePPI has comparable overlap, given that it includes only computational predictions. Because most of the predictions in all databases are unique it seems clear that the information contained in different databases is complementary. A resource combining them would be valuable.

A criticism that has been leveled at many interaction databases is that their contents are biased towards well-studied proteins. *Rolland et al. (2014)* addressed this by ranking proteins in interaction databases according to the number of publications in which they appear; a protein is assumed to be well-studied if it appears in many publications. They demonstrate that the interactions they identify (the Y2H set) overcome this bias. To see if this is also true of PrePPI, we analyzed a set of 3000 proteins with interactions in the Y2H set for which no interactions had been obtained in a in a previous similar study (*Rual et al., 2005*), i.e., proteins with newly identified interactions. For these proteins, PrePPI predicts 327,000 reliable PPIs (LR > 600). In other words, proteins that encompass 15% of the human proteome and are defined as not well-studied are involved in 25% of PrePPI interactions. Moreover 596 of these coincide with interactions in the Y2H set, which accounts for more than half of the total overlap (*Table 2*). These results suggest that PrePPI predictions are relatively immune to the bias towards well-studied proteins. We also note that this is further demonstrated by the fact that PrePPI covers most of the human genome.

## Direct physical interactions

As discussed above, PrePPI makes about reliable 127,000 predictions based only on evidence that indicates a direct interaction (structural modeling – SM; protein peptide – PrP, Protein redundancy – PR). While structural LR ($Max(LR_{SM}, LR_{PrP})*LR_{PR} > 600$ is the most reliable criterion for a direct interaction, structural information contributes to many other reliable predictions and, indeed, ~571,000 predictions have $LR_{PrePPI} > 600$ with the additional criterion that $Max(LR_{SM}, LR_{PrP})*LR_{PR} > 100$, i.e. the LR based on SM, PrP, and PR evidence alone, results in an LR >100. Thus, there may be a much

broader pool of predicted direct interactions than obtained from the stricter criterion of structural LR > 600, possibly on the order of 500,000.

## PrePPI availability and usage

The PrePPI database can be accessed at https://honiglab.c2b2.columbia.edu/PrePPI. Interactions for the human proteome can be downloaded in full, and the GSEA derived functional information and structural models of interactions can be downloaded as well for individual proteins. The statistical tests described above used to evaluate performance suggest that PrePPI should constitute a valuable source of information regarding the interaction, functional and structural properties of a protein. Indeed, PrePPI performance appears comparable to that of experimental databases, as measured by ROC curves, and its coverage is far more extensive. On the other hand, our expectation is that any single PrePPI prediction is less reliable than one obtained from high throughput experimental approaches which are carried out in a cellular environment and reflect, to varying extents, more realistic biological conditions. To provide a more comprehensive resource, the PrePPI server makes available experimentally determined interactions downloaded from various experimentally derived databases. These are also assigned an LR using a Bayesian approach (*Zhang et al., 2013*) which is combined with the PrePPI-derived LR's to assign a final likelihood for an interaction. (Note that this 'database LR' is not used in any of the analyses described here).

The aim of the server is to be a hypothesis-generating tool with the broadest possible scope and coverage. However, the uncertainty associated with an LR means that no prediction can be treated uncritically. In order to facilitate further analyses, PrePPI provides interactive features that include: sorting tools to focus on the predictions that are strongest based on a particular type of evidence (e.g. structural); filters based on different types of functional information such as GO annotation, either taken from UniProt or using the wide range of GSEA predicted annotations available; a filter based on sub-cellular localization; atomic coordinates of structural models which provide testable hypotheses about predicted physical interactions and which provide functional and mechanistic insights not available from binary measures of PPIs.

It should be understood, however, that models used in PrePPI are only approximate, in part because they are created in a coarse-grained and high-throughput manner. They may contain non-physical features such as steric clashes, and there may be isolated cases where a prediction is made using an incorrect model. In general, however, based on previously described experiments where interactions were disrupted by mutating residues in predicted interfaces (*Zhang et al., 2012*) and, as described here, based on the enrichment of disease-associated SNPs and depletion of benign SNPs in interfaces, models should provide a useful approximation of the protein-protein binding site. Nevertheless, it should be appreciated that, as with many other systems biology tools, PrePPI is a starting point for further investigation, not an end point. However its scope, its large-scale use of structural information and its genome-wide functional annotation renders it as a unique resource with broad applicability for biomedical discovery.

## Materials and methods

### PPI reference sets

We compiled several reference sets of PPIs. They are named with one of three classifications: (1) 'HC,' for 'high confidence,' to indicate these interactions are supported by at least two publications; (2) 'experimental' to indicate they were obtained from databases of experimentally determined interactions: DIP (*Salwinski et al., 2004*), MINT (*Chatr-aryamontri et al., 2007*) BioGrid (*Stark et al., 2011*), MIPS (*Mewes et al., 1997*), IntAct (*Kerrien et al., 2007*) and HPRD (*Keshava Prasad et al., 2009*); and (3) 'computational' to indicate the use of computationally derived PPIs from HumanNet (*Franceschini et al., 2013*), PIP (*McDowall et al., 2009*), and OPHID (*Brown and Jurisica, 2007*). The subsets of data and predictions considered in this study are: (1) the 'yeast HC reference' set, containing 42,636 yeast interactions; (2) the 'human HC reference' set, containing 26,983 human interactions; (3) the 'human experimental' set, containing 169,368 human interactions in the union of experimental databases; (4) the 'yeast experimental' set, containing 274,606 yeast interactions in the union of experimental databases; (5) the 'other experimental' set, containing 847,982 interactions from a range of species in the union of experimental databases; (6)

for training over yeast, a negative (N) set contains all pairs for which there is no literature evidence of interaction; (7) for evaluation purposes, the 'human N set' is created randomly choosing 2000 proteins annotated in GO as 'integral to membrane', 500 annotated as 'nucleoplasm', 500 as 'mitochondria' and 383 as 'endoplasmic reticulum' and pairing proteins "integral to membrane" with proteins in 'nucleoplasm', proteins in 'nucleoplasm' with proteins in 'mitochondria' and 'endoplasmic reticulum' and proteins in 'mitochondria' with proteins in 'endoplasmic reticulum', resulting in 1,632,716 pairs (proteins annotated to more than one cellular compartment or contained in the human HC set were excluded) (8) the 'Computational All' set, containing the union of databases of computational predictions; (9) the Y2H set, consisting of 13,584 interactions determined obtained in (*Rolland et al., 2014*); and (10) the BioPlex set consisting of 56,553 obtained in (*Huttlin et al., 2015*). As detailed in the text and below, datasets 1 through 7 were used in training and evaluating the PrePPI Bayesian network. For the ROC curves in *Figure 2*, TPR=TP/$n_{HC}$, where TP is the number true positive predictions in the human HC set with an LR above a given cutoff and $n_{HC}$ is the total number of interactions in the human HC reference set; FPR=FP/$n_N$, where FP is the number of false positive predictions in the N set with an LR above a given cutoff and $n_N$ is the total number of pairs of proteins in the N set.

## PrePPI

The original version of PrePPI has been fully described elsewhere (*Zhang et al., 2012*). The set of proteins for which PPI predictions are made is the human proteome as defined by UniProt (*Consortium 2010*). PrePPI predicts a potential interaction between two given proteins, denoted A and B, by combining different sources of interaction evidence in a Bayesian framework. The evidence includes similarity of their expression profiles, function, phylogenetic history, sequence and structural information.

## Protein structures, template complexes and identification of structurally similar proteins

Three-dimensional models for full-length proteins and protein domains (defined by the Conserved Domain Database [*Marchler-Bauer et al., 2011*]) are either (1) taken directly from the PDB (*Berman et al., 2000*) if experimentally determined structures for the full-length protein or its subdomains exist, or (2) constructed by homology modeling. For homology modeling, the template for a given sequence is identified using the program HHBLITS (*Remmert et al., 2012*) based on having an E-value less than $10^{-12}$. Models were built with MODELLER (*Eswar et al., 2003*) based on the alignment provided by HHBLITS. Overall, a total of 56,351 3D structures, coming from 14,737 PDB structures and 41,614 homology models for at least a portion of ~15,000 proteins, were used in the construction of the current PrePPI database.

Structures of protein complexes used as templates for interaction models are defined either by (1) the PDB 'biounit' files, which are expected to represent the biologically relevant quaternary structure for interacting proteins, (2) the PISA database (*Krissinel and Henrick, 2007*) of computationally predicted biounits, or, if the other two sources are unavailable, (3) the PDB file itself even if uncertainty exists as to whether an interface is biologically relevant. The latter occurred 27,726 times out of the total of 162,833 templates. For each human protein or domain modeled, structurally similar proteins are identified with the program Ska (*Yang and Honig, 2000*, *Petrey and Honig, 2003*) and a protein structural distance less than 0.6 (PSD < 0.6).

## Bayesian network

Scores for a given type of interaction evidence, X, are calculated for every pair of proteins for which the evidence is available. The scores for each type of evidence are partitioned into *n* bins $b_1$, $b_2$,..., $b_n$. In the Bayesian network, a likelihood ratio $LR^X_{b_i}$ is assigned to each bin. $LR^X_{b_i}$ is the percentage of protein pairs in the yeast HC reference set with a score for X in bin $b_i$ divided by the percentage of protein pairs in the yeast N set with a score for X in the same bin $b_i$. Details regarding the scores and associated LRs based on structural modeling (SM) (*Zhang et al., 2012*), protein-peptide (PrP) (*Chen et al., 2015*), gene ontology (GO) (*Jansen et al., 2003*) and phylogenetic profile (PP) (*Zhang et al., 2012*) evidence are described elsewhere. The calculation of scores for the new

evidence based on orthology (OR), RNA expression profile correlation (EP), and partner redundancy (PR) are described below.

The LRs for the different types of evidence (except for SM and PrP) are integrated with a naïve Bayes approach, i.e. by taking their product. The maximum of $LR^{SM}$ and $LR^{PrP}$ is used since the SM- and PrP-derived evidence is based on mutually exclusive interaction models, i.e. two structured domains *versus* a structured domain and a peptide, respectively. The final LR is thus

$$LR_{PrePPI} = \max\left(LR^{SM}, LR^{PrP}\right) \times LR^{PR} \times LR^{GO} \times LR^{PP} \times LR^{OR} \times LR^{EP} \times LR^{PR} \tag{1}$$

## Orthology

The orthology component of the Bayesian network is based on the assumption that if two proteins are known to interact in one species, their orthologs are likely to interact in another (*Matthews et al., 2001*). Detecting orthologous relationships is a complex problem, and a range of methods have been developed with corresponding results compiled into databases, including KEGG (*Kanehisa and Goto, 2000*), EggNOG (*Huerta-Cepas et al., 2016*), Hogenom (*Penel et al., 2009*), OMA (*Altenhoff et al., 2015*), OrthoDB (*Kriventseva et al., 2015*), Ensembl Compara (*Flicek et al., 2014*), Gopher (*Edwards, 2006*) and InParanoid (*Sonnhammer and Östlund, 2015*). In an attempt to select the most informative method, we trained a Bayesian network on each of these eight databases and compared their performance with ROC plots with results shown in *Figure 7*. For this purpose, interactions from other species are identified from the 'other experimental' set described above. Performance varied considerably: Some databases reliably predict orthologous interactions (high TPR at low FPR) for only a few proteins, while others predict orthologous relationships for many more protein pairs but do so unreliably (low TPR compared to FPR). We found that Gopher, OMA, and KEGG exhibit the best performance.

Therefore, rather than define orthologous relationships with a single method or database, the different resources were combined. A four-dimensional vector is defined in which each component can take one of three values: 0, 1, or >1. For a given pair of proteins A and B, these values represent the number of times an ortholog of A and an ortholog of B are found to interact in the 'other experimental' set of interactions described above. The different components of the vector indicate from which database orthologs are obtained: (1) Gopher, (2) OMA, (3) KEGG, or (4) the union of

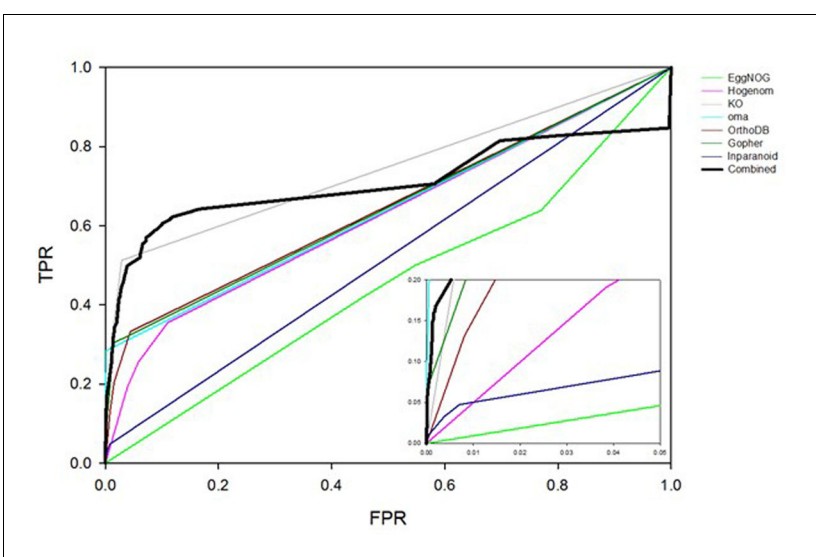

**Figure 7.** ROC plots comparing interaction prediction performance based on orthology for individual databases indicated in the legend. Three bins are defined for any given database, representing whether there 0, 1 or >1 orthologs of two given proteins reported to interact in some other species. LR's for each bin are trained using the yeast HC and N reference sets of interactions. ROC curves are plotted for the human HC and, for this analysis only, a negative reference set consisting of ~200M pairs of proteins for which there is no literature evidence of an interaction.

EggNOG, Hogenom, OrthoDB, Ensmbl Compara, and InParanoid. For example, the vector (1, >1, 0, 0) denotes that orthologs of A and B defined by Gopher interact in one species, orthologs of A and B defined by OMA interact in two or more species, no orthologs of A and B defined by KEGG interact, and no orthologs of A and B taken from the union of the remaining databases interact. Each vector represents a bin for the orthology term. There was insufficient data in the yeast HC reference set to calculate an LR for all possible 81 four-component bins. Therefore, lower-dimensional vectors were also determined by considering a smaller subset of databases. When making a PPI prediction for a given pair of proteins, the highest dimensional vector supported by sufficient data is used to obtain $LR^{OR}$. The Bayesian network based on information from all the databases outperforms the Bayesian networks based on any individual database (see *Figure 7*). Bins and associated LRs for orthology are shown in *Supplementary file 2* ('LRs for Orthology, Partner Redundancy and Expression Profile evidence').

## Partner redundancy

Our approach is a modification of that developed by de Chassey et al (*de Chassey et al., 2008*) in which interactions between virus and host proteins were predicted. Partner Redundancy is based on the extent to which proteins that are structurally similar to A are found to interact with B in the 'human experimental' set of PPIs defined above. More formally, if there are $m_a$ proteins $N_1^A, N_2^A, \ldots, N_{m_a}^A$ in the PDB that are structurally similar to protein A, and $n$ of these proteins are found to interact with B, then the likelihood that A interacts with B increases with $n$. We thus define bins based on the number $n$, and on another property of B, the total number of proteins, $d$, which interact with B in the 'human experimental' set. A bin here is thus a vector of ranges. For example, the bin $([2,5],[0,100])$ means that $n$ is between two and five, and $d$ is between 0 and 100.

A drawback to this approach is that 'known interactions' are defined by membership in the 'human experimental' set and will necessarily involve only human proteins, thus limiting its statistical power. We therefore extended the analysis. When an experimentally determined structure or model of B is available, the following analysis is performed: Bins are defined based on the number of times, $m$, that any $N_i^A$ interacts with any protein $N_1^B, N_2^B, \ldots, N_{m_b}^B$, structurally similar to B. While, this greatly increased the number of pairs of proteins for which an LR is calculated, the evidence was correspondingly weaker since many interactions occur in remotely related species. $LR^{PR}$ is taken from the first Bayesian network if possible and, otherwise, from the second one. Note that both A and B were excluded from $\{N_i^A\}$ and $\{N_i^B\}$. The combination of these two Bayesian networks was found to outperform each individually when incorporated into $LR_{PrePPI}$. Bins and associated LRs for partner redundancy are shown in *Supplementary file 2*.

## Expression profiles

Preprocessed co-expression datasets including pre-calculated pairwise Pearson's correlation coefficients for 11 species (human, nematode, dog, fly, zebrafish, chicken, rhesus monkey, mouse, rat, budding yeast and fission yeast) were collected from COXPRESdb (*Okamura et al., 2015*) and ArrayExpress (*Kolesnikov et al., 2015*) with accession numbers listed in *Table 4.*

Given two human proteins A and B, we find their orthologs $A_j$ and $B_j$ in non-human species j as defined by Gopher because these eleven species are well-represented in this orthology database. From the expression databases listed in *Table 4*, the Pearson correlation coefficients, $c_j$, for the correlated expression of $A_j$ and $B_j$, are used to define the scores $S_{pos}$ and $S_{neg}$ when the correlation coefficients are positive and negative (anti-correlated), respectively, as follows:

$$S_{pos}\left(c_1, c_2, \ldots, c_{n_{pos}}\right) = 1 - \prod_{j=1}^{n_{pos}}\left(1 - c_j\right) \tag{2A}$$

and

$$S_{neg}\left(c_1, c_2, \ldots, c_{n_{neg}}\right) = -\left(1 - \prod_{j=1}^{n_{neg}}\left(1 - |c_j|\right)\right). \tag{2B}$$

These scores reward consistency between different data sources, and similar approaches have been used previously to combine biological evidence in protein function annotation (*Forslund and Sonnhammer, 2008*). The more species j for which $A_j$ and $B_j$ have positive correlation (large $n_{pos}$), the larger $S_{pos}$ will be, and vice versa for $S_{neg}$. The net effect of these formulae is to return a number

**Table 4.** Accession numbers, species, Affymetrix ID and source database for the expression profile datasets used for the expression profile evidence.

| Accession | Species | Affymetrix ID | Source |
|---|---|---|---|
| Hsa.c4-0 | Human | HG-U133_Plus_2 | Coxpressdb |
| Hsa2.c1-0 | Human | HuGene-1_0-st-v1 | Coxpressdb |
| E-MTAB-62 | Human | HG-U133A | ArrayExpress |
| Mmu.c3-0 | Mouse | Mouse430_2 | Coxpressdb |
| Rno.c2-0 | Norway rat | Rat230_2 | Coxpressdb |
| Gga.c2-0 | Chicken | Chicken | Coxpressdb |
| Dre.c2-0 | Zebrafish | Zebrafish | Coxpressdb |
| Dme.c2-0 | Fruit fly | Drosophila_2 | Coxpressdb |
| Cel.c2-0 | Nematoda | Celegans | Coxpressdb |
| Mcc.c1-0 | Rhesus monkey | Rhesus | Coxpressdb |
| Cfa.c1-0 | Dog | Canine_2 | Coxpressdb |
| Sce.c1-0 | Budding yeast | Yeast_2 | Coxpressdb |
| Spo.c1-0 | Fission yeast | Yeast_2 | Coxpressdb |

with absolute value less than 1 but greater than the absolute value of any of the original $c_j$. $S_{orth}$ is then defined as:

$$S_{orth} = \begin{cases} S_{pos} & if\, n_{pos} \geq n_{neg} \\ S_{neg} & if\ n_{pos} < n_{neg} \end{cases} \tag{3}$$

Since there are multiple expression datasets for human (first three rows in *Table 4*), an additional score, $S_{human}$, for proteins $A_j$ and $B_j$ is calculated in a similar way using the correlation coefficients $c_j$ from the human datasets listed above. We then define a cross-species correlation score

$$COXS(A,B) = \begin{cases} 1-(1-S_{human})*(1-S_{orth}*w) & if\ S_{human} \geq 0\ and\ S_{orth} \geq 0 \\ S_{human} & if\ S_{human} \geq 0\ and\ S_{orth} < 0 \\ S_{orth} & if\ S_{human} < 0\ and\ S_{orth} \geq 0 \\ -(1-(1-|S_{human}|)*(1-|S_{orth}|*w)) & if\ S_{human} < 0\ and\ S_{orth} < 0 \end{cases} \tag{4}$$

where $w$=0.6 which was found to give the best performance by varying $w$ and comparing COXS-based ROC curves for the recovery of the human HC reference set. Bins and associated LRs for expression profiles are listed in *Supplementary file 2*.

## Gene set enrichment analysis

Gene set enrichment analysis was carried out with software downloaded from the mSigDB (*Subramanian et al., 2005*) web site using the 'C5' collection of gene sets associated with GO terms. Gene sets were considered enriched if the q-value (false discovery rate) reported by the software was less than 0.01. The average q-value for the top 10 enriched gene sets was 0.0002.

## SNP analysis

To determine if SNPs were enriched in protein interaction interfaces, we compiled: (1) a set of 73,552 SNPs in ~3,800 proteins (from ClinVar and the 1000 Genomes Project) that were annotated as disease-associated using data from the Human Gene Mutation Database (*Stenson et al., 2014*) and (2) a set of 68,210 SNPS annotated as benign in 14,000 proteins from the 1000 Genomes Project. For each set, we find all reliable (LR>600) PrePPI predictions involving the mutated proteins with interaction models. We count the total number of interfacial/non-interfacial residues in the proteins containing the SNPs as well as the number of interfacial/non-interfacial SNPs, obtaining the data in *Table 5*. For both the disease-associated and benign sets, $n_{11}$ is the number of SNPs that are

**Table 5.** Contingency tables for analyzing enrichment of SNPs in modeled interfaces. For both the disease-associated and benign sets, the numbers in parentheses are as follows: n11 is the number of SNPs that are interfacial, n10 is the number that are not interfacial, n01 is the number of interfacial unmutated residues and n00 is the number of non-interfacial, unmutated residues. These numbers are used in the odds-ratio and Z-score calculation as described in the 'SNP Analysis' section.

| | Disease | |
| --- | --- | --- |
| | Interfacial | Non-interfacial |
| Mutated | 12,151 ($n_{11}$) | 61,401 ($n_{10}$) |
| Non-mutated | 298,442 ($n_{01}$) | 2,387,146 ($n_{00}$) |
| | Benign | |
| Mutated | 3554 ($n_{11}$) | 64,656 ($n_{10}$) |
| Non-mutated | 845,876 ($n_{01}$) | 8,221,708 ($n_{00}$) |

interfacial, $n_{10}$ is the number that are not interfacial, $n_{01}$ is the number of interfacial unmutated residues and $n_{00}$ is the number of non-interfacial unmutated residues. The odds ratio is calculated as $OR = \frac{p_1/(1-p_1)}{p_2/(1-p_2)}$. Here $p_1 = n_{11}/(n_{11} + n_{10})$, is the fraction of SNPs that are interfacial, and $p_2 = (n_{11} + n_{10})/(n_{11} + n_{10} + n_{01} + n_{00})$ is the fraction of all residues that are interfacial. The log-odds standard error was calculated as $SE_{log-odds} = \sqrt{\frac{1}{n_{11}} + \frac{1}{n_{10}} + \frac{1}{n_{01}} + \frac{1}{n_{00}}}$, and a Z-score calculated as $Z = \frac{\ln(OR)}{SE_{log-odds}}$. The Z-score for the enrichment of disease associated SNPs in interfaces was ~41 and for the depletion of benign SNPs was ~35, implying the odds ratios we find are statistically significant.

## Acknowledgements

This work was funded by NIH grant GM030518 to BH and equipment grants S10OD012351 and S10OD021764 to the Department of Systems Biology.

## Additional information

### Funding

| Funder | Grant reference number | Author |
| --- | --- | --- |
| National Institutes of Health | GM030518 | Barry Honig |
| National Institutes of Health | S10OD012351 | José Ignacio Garzón<br>Lei Deng<br>Diana Murray<br>Sagi Shapira<br>Donald Petrey<br>Barry Honig |
| National Institutes of Health | S10OD021764 | José Ignacio Garzón<br>Lei Deng<br>Diana Murray<br>Sagi Shapira<br>Donald Petrey<br>Barry Honig |

The funders had no role in study design, data collection and interpretation, or the decision to submit the work for publication.

### Author contributions

JIG, LD, DP, Conception and design, Acquisition of data, Analysis and interpretation of data, Drafting or revising the article; DM, Analysis and interpretation of data, Drafting or revising the article; SS, Conception and design, Analysis and interpretation of data; BH, Conception and design, Analysis and interpretation of data, Drafting or revising the article

## Author ORCIDs
Barry Honig, http://orcid.org/0000-0002-2480-6696

# Additional files

## Supplementary files
• Supplementary file 1. UniProt codes for interacting proteins that are related to the same disease. Each line shows a pair of proteins annotated as associated with the same disease in the ClinVar database.

• Supplementary file 2. LRs for orthology, partner redundancy and expression profile interaction evidence.

## Major datasets
The following previously published datasets were used:

| Author(s) | Year | Dataset title | Dataset URL | Database, license, and accessibility information |
|---|---|---|---|---|
| Salwinski L, Miller CS, Smith AJ, Pettit FK, Bowie JU, Eisenberg D | 2004 | The Database of Interacting Proteins: | http://dip.doe-mbi.ucla.edu/dip/Download.cgi?SM=10 | Available at the Database of Interacting Proteins (http://dip.mbi.ucla.edu/dip/) |
| Chatr-Aryamontri A, Ceol A, Palazzi LM, Nardelli G, Schneider MV, Castagnoli L, Cesareni G | 2007 | MINT: the Molecular INTeraction database | ftp://ftp.ebi.ac.uk/pub/databases/intact/current/all.zip | Available at the IntAct website (http://www.ebi.ac.uk/intact/) |
| Stark C, Breitkreutz BJ, Chatr-Aryamontri A, Boucher L, Oughtred R, Livstone MS, Nixon J, Van Auken K, Wang X, Shi X, Reguly T, Rust JM, Winter A, Dolinski K, Tyers M | 2011 | The BioGRID Interaction Database | https://thebiogrid.org/downloads/archives/Release%20Archive/BIOGRID-3.4.141/BIOGRID-ALL-3.4.141.tab.zip | Available at the BioGRID General Repository (https://thebiogrid.org/) |
| Mewes HW, Albermann K, Heumann K, Liebl S, Pfeiffer F | 1997 | MIPS | http://mips.helmholtz-muenchen.de/proj/ppi/data/mppi.gz | Available at the MIPS Mammalian Protein-Protein Interaction Database (http://mips.helmholtz-muenchen.de/proj/ppi/) |

| | | | | |
|---|---|---|---|---|
| Keshava Prasad TS, Goel R, Kandasamy K, Keerthikumar S, Kumar S, Mathivanan S, Telikicherla D, Raju R, Shafreen B, Venugopal A, Balakrishnan L, Marimuthu A, Banerjee S, Somanathan DS, Sebastian A, Rani S, Ray S, Harrys Kishore CJ, Kanth S, Ahmed M, Kashyap MK, Mohmood R, Ramachandra YL, Krishna V, Rahiman BA, Mohan S, Ranganathan P, Ramabadran S, Chaerkady R, Pandey A | 2009 | Human Protein Reference Database | http://hprd.org/download | Available at Human Protein Reference Database (http://hprd.org/download) |
| Franceschini A, Szklarczyk D, Frankild S, Kuhn M, Simonovic M, Roth A, Lin J, Minguez P, Bork P, von Mering C, Jensen LJ | 2013 | HumanNet | http://www.functionalnet.org/humannet/HumanNet.v1.join.txt | Available at HumanNet (http://www.functionalnet.org/humannet/about.html) |
| McDowall MD, Scott MS, Barton GJ | 2009 | PIPs | http://www.compbio.dundee.ac.uk/www-pips/downloads/PredictedInteractions100.txt | Available at Human Protein-Protein Interaction Prediction website (http://www.compbio.dundee.ac.uk/www-pips/) |
| Brown KR1, Jurisica I | 2007 | OPHID | http://ophid.utoronto.ca/ophidv2.204/downloads.jsp | Available at I2D Interologous Interaction Database (http://ophid.utoronto.ca/ophidv2.204/index.jsp) |
| Kanehisa M, Goto S | 2000 | KEGG | http://www.genome.jp/kegg-bin/download_htext?htext=ko00001.keg&format=htext&filedir= | Available at KEGG PATHWAY Database (http://www.genome.jp/kegg/) |
| Huerta-Cepas J, Szklarczyk D, Forslund K, Cook H, Heller D, Walter MC, Rattei T, Mende DR, Sunagawa S, Kuhn M, Jensen LJ, von Mering C, Bork P | 2016 | eggNOG | http://eggnog.embl.de/version_3.0/data/downloads/all.members.tar.gz | Available at eggNOG database (http://eggnogdb.embl.de/#/app/home) |
| Penel S, Arigon AM, Dufayard JF, Sertier AS, Daubin V, Duret L, Gouy M, Perrière G | 2009 | Hogenom | ftp://pbil.univ-lyon1.fr/pub/hogenom/release_06/hogenom6_prot.tar.gz | Available at Hogenom database (ftp://pbil.univ-lyon1.fr/pub/hogenom/) |
| Altenhoff AM, Škunca N, Glover N, Train CM, Sueki A, Piližota I, Gori K, Tomiczek B, Müller S, Redestig H, Gonnet GH, Dessimoz C | 2015 | OMA | http://omabrowser.org/All/oma-pairs.txt.gz | Available at OMA Orthology database (http://omabrowser.org/oma/home/) |

| | | | | |
|---|---|---|---|---|
| Kriventseva EV, Te-genfeldt F, Petty TJ, Waterhouse RM, Simão FA, Pozdnyakov IA, Ioannidis P, Zdobnov EM | 2015 | OrthoDB | http://www.orthodb.org/v9/download/odb9_OGs.tab.gz | Available at OrthoDB database (http://www.orthodb.org/) |
| Okamura Y, Aoki Y, Obayashi T, Tadaka S, Ito S, Narise T, Kinoshita K | 2015 | Coexpression data for C. elegans | http://coxpresdb.jp/download/Cel.v12-08.G17256-S1034.rma.mrgeo.d.tar.bz2 | Publicly available at Coexpressdb (accession no: Cel.c2-0). |
| Okamura Y, Aoki Y, Obayashi T, Tadaka S, Ito S, Narise T, Kinoshita K | 2015 | Coexpression data for dog | http://coxpresdb.jp/download/Cfa.v12-08.G16211-S377.rma.mrgeo.d.tar.bz2 | Publicly available at Coexpressdb (accession no: Cfa.c1-0). |
| Okamura Y, Aoki Y, Obayashi T, Tadaka S, Ito S, Narise T, Kinoshita K | 2015 | Coexpression data for fruit fly | http://coxpresdb.jp/download/Dme.v12-08.G12626-S3336.rma.mrgeo.d.tar.bz2 | Publicly available at Coexpressdb (accession no: Dme.c2-0). |
| Okamura Y, Aoki Y, Obayashi T, Tadaka S, Ito S, Narise T, Kinoshita K | 2015 | Coexpression data for zebrafish | http://coxpresdb.jp/download/Dre.v12-08.G10112-S1126.rma.mrgeo.d.tar.bz2 | Publicly available at Coexpressdb (accession no: Dre.c2-0). |
| Okamura Y, Aoki Y, Obayashi T, Tadaka S, Ito S, Narise T, Kinoshita K | 2015 | Coexpression data for chicken | http://coxpresdb.jp/download/Gga.v12-08.G13757-S1024.rma.mrgeo.d.tar.bz2 | Publicly available at Coexpressdb (accession no: Gga.c2-0). |
| Okamura Y, Aoki Y, Obayashi T, Tadaka S, Ito S, Narise T, Kinoshita K | 2015 | Coexpression data for human | http://coxpresdb.jp/download/Hsa.v12-08.G19803-S73083.rma.mrgeo.d.tar.bz2 | Publicly available at Coexpressdb (accession no: Hsa.c4-0). |
| Okamura Y, Aoki Y, Obayashi T, Tadaka S, Ito S, Narise T, Kinoshita K | 2015 | Coexpression data for human | http://coxpresdb.jp/download/hsa.v14-04.G19816-S5626.quantile.mrgeo.d.tar.bz2 | Publicly available at Coexpressdb (accession no: Hsa2.c1-0). |
| Okamura Y, Aoki Y, Obayashi T, Tadaka S, Ito S, Narise T, Kinoshita K | 2015 | Coexpression data for rhesus monkey | http://coxpresdb.jp/download/Mcc.v12-08.G15779-S675.rma.mrgeo.d.tar.bz2 | Publicly available at Coexpressdb (accession no: Mcc.c1-0). |
| Okamura Y, Aoki Y, Obayashi T, Tadaka S, Ito S, Narise T, Kinoshita K | 2015 | Coexpression data for mouse | http://coxpresdb.jp/download/Mmu.v12-08.G20403-S31479.rma.mrgeo.d.tar.bz2 | Publicly available at Coexpressdb (accession no: Mmu.c3-0). |
| Okamura Y, Aoki Y, Obayashi T, Tadaka S, Ito S, Narise T, Kinoshita K | 2015 | Coexpression data for Norway rat | http://coxpresdb.jp/download/Rno.c2-0.G13751-S3526.tar.gz | Publicly available at Coexpressdb (accession no: Rno.c2-0). |
| Okamura Y, Aoki Y, Obayashi T, Tadaka S, Ito S, Narise T, Kinoshita K | 2015 | Coexpression data for budding yeast | http://coxpresdb.jp/download/Sce.v12-08.G4461-S2693.rma.mrgeo.d.tar.bz2 | Publicly available at Coexpressdb (accession no: Sce.c1-0). |
| Okamura Y, Aoki Y, Obayashi T, Tadaka S, Ito S, Narise T, Kinoshita K | 2015 | Coexpression data for fission yeast | http://coxpresdb.jp/download/Spo.v14-08.G4881-S224.rma.mrgeo.d.tar.bz2 | Publicly available at Coexpressdb (accession no: Spo.c1-0). |

| Kolesnikov N, Hastings E, Keays M, Melnichuk O, Tang YA, Williams E, Dylag M, Kurbatova N, Brandizi M, Burdett T, Megy K, Pilicheva E, Rustici G, Tikhonov A, Parkinson H, Petryszak R, Sarkans U, Brazma A | 2015 | Coexpression data for human | https://www.ebi.ac.uk/arrayexpress/files/E-MTAB-62/E-MTAB-62.processed.2.zip | Publicly available at the EMBL-EBI Array Express database (https://www.ebi.ac.uk/arrayexpress/) |

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
