## [Decision Letter]

Thank you for submitting your article "A computational interactome and functional annotation for the human proteome" for consideration by *eLife*. Your article has been favorably evaluated by Naama Barkai as the Senior Editor and three reviewers, one of whom, Nir Ben-Tal (Reviewer #1), is a member of our Board of Reviewing Editors.

The reviewers have discussed the reviews with one another and the Reviewing Editor has drafted this decision to help you prepare a revised submission.

Summary:

The Honig lab has been developing the PrePPI methodology and web-server/database for predicting protein-protein interactions at genome-wide scale. PrePPI infers interactions based on known protein-protein complexes, as well as direct and indirect experimental techniques, such as phylogenetic profiles, gene ontology, expression profiles, etc. Here they present a revised PrePPI which is shown to be much superior to the previous version in terms of coverage and accuracy. Improvement is based on data accumulation, which is trivial (but very important!), but also on the addition of novel characteristics of protein-protein interactions, such as information from orthologues, expression profiles, partner redundancy and protein/peptide interactions. PrePPI now covers more than 1.35M interactions in the human proteome, 4 times more than the original version. The paper additionally reports the use of this information for function annotation, gene set enrichment analysis (GSEA), and an analysis of SNPs at interfaces. This is a substantial piece of research that merits publication in *eLife* if the following points can be addressed.

Essential revisions:

A) The algorithm

1) Features used for scoring the interaction likelihood: How is "5- Orthology" different from "2- Phylogenetic-Profile"?

2) Features used for scoring the interaction likelihood: How is Expression Profile used differently in the new vs. old PrePPI?

3) How specific are the predicted interactions? Considering known interaction pairs of orthologs from two protein families A and B, can PrePPI correctly pair A1 to B1, A2 to B2 and so on?

4) Is it the case that the vast majority of predicted interactions are among paralogous proteins, where missing interactions are inferred based on concrete data for a pair of interacting proteins? It would be helpful to report the fraction of these.

B) Evaluation of coverage and accuracy

5) Training on yeast and testing with human: What is the cross-sets similarity between the proteins in both sets in terms of sequence and structure? Because some of the descriptors are based also on orthologues, how similar are the proteins in the sets in terms of their representations?

6) The authors include in the definition of PPI proteins that are functionally related. This is a very generous interpretation of a PPI which is helpful for function annotation and GSEA. The Abstract should clearly state the definition of PPI and note that there are about 50,000 reliable predictions based on structural evidence alone which is indicative of a direct interaction (see subsection “Prediction Performance”, last paragraph).

7) To get an estimate of the expected accuracy, it could be insightful to compare the new and old PrePPI. The manuscript gives an overall comparison in terms of counts of interactions etc., but comparison of individual interactions would complete the picture. For example, is the difference between the old and new PrePPI only in better coverage? What fraction of the interactions that have been assigned high confidence in the old prePPI are dismissed in the new prePPI? For these (if found), what are the reasons? Changes in the available structural or non-structural data? Or is it the result of smarter data processing?

8) When querying the web site with human trypsin type 1 (P07477), one of the interactors was P35030 (human trypsin type 3). Strangely the structural model for the interaction showed the two trypsin structures were superposed. What does this mean? It clearly is not biologically sensible. This raised the question of whether the website and the entire approach have been stress tested by detailed analysis of specific examples rather than global metrics (ROC curves etc.). The authors should generate a random list of a few Uniprot IDs and input them to the server and then carefully check the answers. This requires work by a researcher with extensive biological expertise – so odd predictions can be identified. The paper could report the set of tested Uniprot IDs.

9) Related to (8) above, please also address the realism of the predicted physical interactions. The original paper acknowledged that the structural models are not refined and are often non-physical. Indeed, often the interacting partners in PrePPI models suffer from major steric clashes. On the other hand, some models are based on non-interacting chains, although they might have high prediction scores.

C) Comparison to other algorithms

10) PrePPI should be compared to other methods. For clear comparison it is essential to clarify the cross-validation since the PrePPI LR-score combines experimental and computational evidence. It is not clear from the paper if experimental data used for training were excluded in testing (Figure 1). This is especially important since PPI databases overlap with each other. How different would the Precision-Recall curves be if only crystal structures with other experimental evidence were used for prediction? In other words, what is the contribution of structural models to the quality of predictions?

11) When compared with the human data from Y2H from Rolland et al., Figure 1 – the authors say that PrePPI compares favorably with Y2H. Please verify and demonstrate this result. Precision-recall curves in addition to ROC are needed in the main text since the sizes of TP and FP datasets are not balanced. What is the size of "N" dataset – millions or hundreds of millions of interactions?

12) The point in (5) is also relevant for the comparison of the third (smaller) set from Vidal to the two other sets.

13) Given the broad definition of a PPI (see 6 above), when PrePPI is compared to the coverage of other databases it would be important to state that these other databases might be using a far more restricted definition of a PPI.

14) The paper describes (subsection “Shared GO annotation”) the number of predictions obtained after removing the GO terms. If it is tractable, it would be interesting to have a table showing the effect of removing each of the terms in turn.

D) Update

15) At present, PrePPI is presumably up-to-date in terms of usage of all experimental data (structure-based or otherwise), and its enhanced performance in comparison to other resources is, in part, related to that. It would be great if PrePPI will be kept up-to-date automatically. It should be possible with the appropriate scripts. Anyway, please indicate in the main text how frequently you plan to update PrePPI.

E) Data availability

16) The web site works for a single sequence input. It is not clear how much global data is available to the community. Can one download a list of all predicted PPIs with the scores? Can one obtain predicted protein-protein structures? This needs to be stated in the paper. According to *eLife* policy *all* the data should be made easily accessible to the public. Please take care of that.

---

## [Author Response]

*[…] Essential revisions:*

*A) The algorithm*

*1) Features used for scoring the interaction likelihood: How is "5- Orthology" different from "2- Phylogenetic-Profile"?*

For proteins A and B, the “orthology” score directly reflects whether the orthologs of A and B are thought to interact in some other species. “Phylogenetic profile” only reflects whether orthologs of A and B are in a similar set of species without explicitly using any information regarding whether they interact. The difference has been clarified in the manuscript (subsection “Overview of the PrePPI algorithm”). In response to this point as well as point 2 below, we have also added a new figure (Figure 1 in the current manuscript) to provide a broad visual overview of the different sources of evidence and how they are used.

2) Features used for scoring the interaction likelihood: How is Expression Profile used differently in the new vs. old PrePPI?

In the old version of “Expression Profile”, we only consider correlated expression of human proteins A and B. In the new version we also incorporate expression information for orthologs of A and B in 11 model species. We have modified the manuscript to highlight this difference (subsection “Overview of the PrePPI algorithm”).

*3) How specific are the predicted interactions? Considering known interaction pairs of orthologs from two protein families A and B, can PrePPI correctly pair A1 to B1, A2 to B2 and so on?*

We now devote a separate section to this comment and the next one (“b) The role of paralogs and related issues of specificity”). Briefly, we show that the structural LR contains some degree of specificity which is then enhanced by non-structural information. We gave specific examples of this in supplemental figures in our previous paper but here we have presented a more general analysis. Perhaps surprisingly, although now rationalized in the manuscript, PrePPI predicts a very wide range of LRs for pairs of proteins that are annotated as paralogs of proteins in the template used for prediction, with higher LRs corresponding to a higher probability of being a known interaction (as determined by current databases).

*4) Is it the case that the vast majority of predicted interactions are among paralogous proteins, where missing interactions are inferred based on concrete data for a pair of interacting proteins? It would be helpful to report the fraction of these.*

This is now discussed in a new section on paralogs (“b) The role of paralogs and related issues of specificity”). In short, about 25% of our reliable structure-based predictions involve paralogs and even among these, there is a considerable degree of specificity (and see response to point 3).

*B) Evaluation of coverage and accuracy*

*5) Training on yeast and testing with human: What is the cross-sets similarity between the proteins in both sets in terms of sequence and structure? Because some of the descriptors are based also on orthologues, how similar are the proteins in the sets in terms of their representations?*

We have now devoted a small section to this question (“a) Independence of training and test sets”). There are 2,671 interactions in the yeast training set where the two proteins also have orthologs (as annotated in Gopher, OMA and KEGG) that interact in the human testing set. To see if this was artificially inflating PrePPI performance, we recalculated ROC plots where LRs were calculated excluding the evidence from orthology for these 2,671 interactions and found essentially identical performance.

*6) The authors include in the definition of PPI proteins that are functionally related. This is a very generous interpretation of a PPI which is helpful for function annotation and GSEA. The Abstract should clearly state the definition of PPI and note that there are about 50,000 reliable predictions based on structural evidence alone which is indicative of a direct interaction (see subsection “Prediction Performance”, last paragraph).*

The Abstract has been modified according to the reviewer’s suggestion where we now say that between 127,000 and 500,000 interaction predictions may be direct. This is based on a reevaluation the number of direct interactions with the major change in estimate due to the addition of partner redundancy information in the current draft and to our highlighting the possibility of there being a much larger number of direct interactions which results from the use of a less stringent criterion. A section in the Discussion is now devoted to this general issue.

We also point out that other large computationally derived databases also use a generous definition of a PPI (subsection “Overlap between databases”, first paragraph). The reviewers clearly appreciated that this definition is very useful in the use of GSEA and we have now added a comment about this point in the Discussion that did not appear in the previous version (see aforementioned paragraph).

*7) To get an estimate of the expected accuracy, it could be insightful to compare the new and old PrePPI. The manuscript gives an overall comparison in terms of counts of interactions etc., but comparison of individual interactions would complete the picture. For example, is the difference between the old and new PrePPI only in better coverage? What fraction of the interactions that have been assigned high confidence in the old prePPI are dismissed in the new prePPI? For these (if found), what are the reasons? Changes in the available structural or non-structural data? Or is it the result of smarter data processing?*

We have now devoted a new section (“c) Overlap between the current and previous versions of PrePPI”) to a comparison between new and old PrePPI. One major difference is that we trained the new Bayesian network on a yeast reference set of interactions that is 4 times larger than the one previously used. This resulted in a global change in the LRs so it would be difficult to point to a single systematic cause of differences in the predictions, although we did find that the LR’s based on structural modeling (SM) were systematically lower. Overall, 37% of reliable interactions in the previous version of PrePPI no longer appear as reliable in the current version. However, this very likely reflects improved prediction accuracy in the new version. This is suggested, as now discussed in the paper, by the fact that nearly all true interactions (in the HC set) that were previously found with LR>600 are assigned an LR>600 in the current version. We also note that ~ 400,000 reliable predictions can be made based on the old evidence in PrePPI, so while better coverage may be contributing to improved performance, so is better data processing in the form of both better training and new sources of evidence.

*8) When querying the web site with human trypsin type 1 (P07477), one of the interactors was P35030 (human trypsin type 3). Strangely the structural model for the interaction showed the two trypsin structures were superposed. What does this mean? It clearly is not biologically sensible. This raised the question of whether the website and the entire approach have been stress tested by detailed analysis of specific examples rather than global metrics (ROC curves etc.). The authors should generate a random list of a few Uniprot IDs and input them to the server and then carefully check the answers. This requires work by a researcher with extensive biological expertise – so odd predictions can be identified. The paper could report the set of tested Uniprot IDs.*

The trypsin prediction is, of course, nonsensical and is an unfortunate drawback to the coarse modeling used in PrePPI. It occurs because trypsin is cleaved and in the PDB (PDB code 1aks), the N and C terminal fragments are labeled as chain A and chain B. PrePPI naturally treats this as two interacting chains and thus a suitable template to model an interaction between trypsin 1 and trypsin 3.

We randomly chose 6 proteins (with UniProt ids/common name: P08069/IGFR1, P35240/Merlin, P63165/SUMO1, Q8WUI4/HDAC, P27348/14-3-3theta and P19174/PLC gamma1) and included, in addition, 3 isoforms of RAS (Uniprot ids P01116, P01112, P01111) which we are examining in detail in other projects. We looked at the top interaction models based on the SM score and found no pathological interaction models like the trypsin example above. We now alert the user to the possibility of encountering incorrect models in a new section “PrePPI availability and usage”. It is difficult, if not impossible, to eliminate a priori any templates that lead to predictions such as the one for trypsin above, and applying filters based on steric clashes would be too computationally cumbersome given the number of models we make. However, we will implement a system where certain PDB files can be excluded forever as templates if strange predictions are reported to us.

If the reviewers agree, we prefer not to add specific examples of results based on our stress testing to the paper since, even though six were chosen randomly, they would almost certainly raise questions about “cherry picking”. But we have now included a few of them as working examples on the server and, in addition, the server will contain appropriate warnings.

*9) Related to (8) above, please also address the realism of the predicted physical interactions. The original paper acknowledged that the structural models are not refined and are often non-physical. Indeed, often the interacting partners in PrePPI models suffer from major steric clashes. On the other hand, some models are based on non-interacting chains, although they might have high prediction scores.*

As discussed above with regard to point 8, the issue of interaction models with steric clashes is now discussed in the section “PrePPI availability and usage”. We did find a few examples of predictions between non-interacting chains in the template in the 9 examples we looked at. In these cases, there was a problem with the way we constructed the models. As it is currently implemented, we always use the PDB file from RCSB as the template to reconstruct the interaction model, but the actual template used to calculate the score may have a different quaternary structure (for example, if the PISA database says that the biologically relevant quaternary structure is not the quaternary structure in the PDB file submitted to RCSB). We are working to fix this, but in the meantime, these instances are not displayed in the interaction model window; rather, a message appears stating that a model could not be built.

*C) Comparison to other algorithms*

*10) PrePPI should be compared to other methods. For clear comparison it is essential to clarify the cross-validation since the PrePPI LR-score combines experimental and computational evidence. It is not clear from the paper if experimental data used for training were excluded in testing (Figure 1). This is especially important since PPI databases overlap with each other. How different would the Precision-Recall curves be if only crystal structures with other experimental evidence were used for prediction? In other words, what is the contribution of structural models to the quality of predictions?*

A comparison to other databases is difficult since the widely used databases typically also serve as repositories for experimentally determined interactions, so their “predictions” will always be in our HC reference set. Note that experimental evidence is not used for any of the analyses in the manuscript. As now pointed out, this data is included on the web site to make PrePPI as comprehensive as possible. We now say this explicitly (subsection “PrePPI availability and usage”, first paragraph) but we feel that given these issues, the comparison to experimental approaches we provide in the manuscript is sufficient to validate PrePPI performance.

We now devote a separate section to homology models which supplements the analysis in our previous manuscript. We show using ROC curves (Figure 2) that models improve performance. However, their most important effect is to increase the number of reliable predictions which cannot be seen in the ROC curves. Only ~250,000 reliable predictions are made without homology models, compared to 1.24 million reliable predictions that include structural evidence if homology models are used.

*11) When compared with the human data from Y2H from Rolland et al., Figure 1 – the authors say that PrePPI compares favorably with Y2H. Please verify and demonstrate this result. Precision-recall curves in addition to ROC are needed in the main text since the sizes of TP and FP datasets are not balanced. What is the size of "N" dataset – millions or hundreds of millions of interactions?*

PrePPI performs poorly in a precision recall curve since it makes so many untested predictions which appear as false positives. For this reason, there is little meaning to the definition of precision. We will of course add these curves to the manuscript if the reviewers disagree. We believe that the ROC curves for the PRS/RRS data sets are meaningful and have changed the text to indicate that PrePPI performs comparably to other databases as measured in this way but has greater coverage. Moreover, we now say explicitly in the Discussion that we expect PrePPI predictions to be less reliable than HT experiments. We have also removed any comparison from the abstract so that the reader will have access to the basis of the comparison in the body of the text.

We have extended our analysis by recalculating the ROC plots using a new N set that was defined based on the criterion that the protein pair are annotated by GO as being in different cellular compartments (see Materials and methods). This was necessary since our previous N set specifically excluded all interactions found by high-throughput experiments, artificially giving the Rolland Y2H and BioPlex a false positive rate of zero. The size of the new N set is 1,632,716 (compared to the N set of size ~200,000,000 in the previous submission) and is now reported in the manuscript. The new N set is smaller and thus permits greater balance than the N set used in the previous submission while yielding similar results. The results shown in the new Figure 2 support our contention that PrePPI performs “comparably” to high throughput databases using this measure but, again, we are very explicit in the discussion about the relative reliability of both approaches.

*12) The point in (5) is also relevant for the comparison of the third (smaller) set from Vidal to the two other sets.*

As in our response to point 5 above, excluding evidence from orthology does not change the ROC plots, so orthology is not artificially inflating PrePPI performance with respect to recovery of the Vidal PRS set.

*13) Given the broad definition of a PPI (see 6 above), when PrePPI is compared to the coverage of other databases it would be important to state that these other databases might be using a far more restricted definition of a PPI.*

We actually explicitly say that the large size of PrePPI may be due to this general definition of a PPI (subsection “Overlap between databases”, first paragraph). We also note that the same definition is implicitly used in other databases.

*14) The paper describes (subsection “Shared GO annotation”) the number of predictions obtained after removing the GO terms. If it is tractable, it would be interesting to have a table showing the effect of removing each of the terms in turn.*

This is now included in Table 1 in the revised manuscript along with an expanded discussion of the effect of the individual sources of evidence (subsection “The contribution of different sources of information to PrePPI performance”).

*D) Update*

*15) At present, PrePPI is presumably up-to-date in terms of usage of all experimental data (structure-based or otherwise), and its enhanced performance in comparison to other resources is, in part, related to that. It would be great if PrePPI will be kept up-to-date automatically. It should be possible with the appropriate scripts. Anyway, please indicate in the main text how frequently you plan to update PrePPI.*

This is now included in a new section of the Discussion called “Assessing PrePPI testing and training”..

*E) Data availability*

*16) The web site works for a single sequence input. It is not clear how much global data is available to the community. Can one download a list of all predicted PPIs with the scores? Can one obtain predicted protein-protein structures? This needs to be stated in the paper. According to eLife policy all the data should be made easily accessible to the public. Please take care of that.*

This is now included in a new section of the Discussion called “PrePPI availability and usage”.